# Endothelial LRP1 regulates metabolic responses by acting as a co-activator of PPARγ

Hua Mao[1], Pamela Lockyer[2], Luge Li[1], Christie M. Ballantyne[3], Cam Patterson[4], Liang Xie[1] & Xinchun Pi[1]

Low-density lipoprotein receptor-related protein 1 (LRP1) regulates lipid and glucose metabolism in liver and adipose tissue. It is also involved in central nervous system regulation of food intake and leptin signalling. Here we demonstrate that endothelial Lrp1 regulates systemic energy homeostasis. Mice with endothelial-specific Lrp1 deletion display improved glucose sensitivity and lipid profiles combined with increased oxygen consumption during high-fat-diet-induced obesity. We show that the intracellular domain of Lrp1 interacts with the nuclear receptor Pparγ, a central regulator of lipid and glucose metabolism, acting as its transcriptional co-activator in endothelial cells. Therefore, Lrp1 not only acts as an endocytic receptor but also directly participates in gene transcription. Our findings indicate an underappreciated functional role of endothelium in maintaining systemic energy homeostasis.

[1] Cardiovascular Research Institute and Department of Medicine, Baylor College of Medicine, Houston, Texas 77030, USA. [2] Department of Pathology and Laboratory Medicine, University of North Carolina at Chapel Hill, Chapel Hill, North Carolina 27599, USA. [3] Center for Cardiovascular Disease Prevention, Methodist DeBakey Heart and Vascular Center, Section of Cardiovascular Research, Department of Medicine, Baylor College of Medicine, Houston, Texas 77030, USA. [4] New York-Presbyterian Hospital, New York, New York 10065, USA. Correspondence and requests for materials should be addressed to X.P. (email: xpi@bcm.edu).

ow-density lipoprotein (LDL)-related protein 1 (LRP1), a multifunctional member of LDL receptor family, is involved in a variety of biological processes, such as lipid metabolism, endocytosis and signal transduction[1–3]. Global deletion of Lrp1 gene in mice leads to embryonic lethality[4]. Depletion of Lrp1 in the livers of LDL receptor-deficient mice results in the accumulation of cholesterol-rich remnant lipoproteins in the circulation, suggesting the critical role of Lrp1 in the clearance of cholesterol-rich remnant lipoproteins[5]. Mice with hepatocyte-specific deletion of Lrp1 also display defects in cholesterol efflux and high-density lipoprotein (HDL) secretion[6]. On the other hand, inactivation of adipocyte Lrp1 results in delayed postprandial lipid clearance, reduced body weight, alterations in adipocyte tissue metabolism and resistance of high-fat-induced glucose tolerance and obesity[7]. The studies of Lrp1 forebrain knockout mice suggest that neuronal Lrp1 regulates food intake and energy homeostasis by directly affecting leptin signalling[8]. Although these tissue-specific knockout studies have provided us enormous information of Lrp1 functions and linked it to lipid metabolism, glucose homeostasis and obesity, the underlying mechanisms remain elusive.

Recently, we and others have demonstrated that Lrp1 is also expressed in endothelial cells (ECs) and can be induced by hypoxia and statins[9–15]. However, whether and how Lrp1 in endothelium regulates energy homeostasis has not been previously studied. Given that lipoprotein receptors play important roles in lipid transport, it is not surprising that endothelial Lrp1 regulates lipid uptake or efflux. In addition, Lrp1 is an endocytic receptor or co-receptor of many ligands such as apolipoprotein E (ApoE) and regulates multiple signalling pathways, such as leptin signalling[8,16,17]. Therefore, we hypothesized that Lrp1 in endothelium might regulate systemic metabolic responses. To study this, we established an EC-specific Lrp1-deficient mouse model, generated by breeding Lrp1[flox/flox] (Lrp1[f/f]) and Tie2Cre[+] transgenic mice followed by bone marrow transplantation (BMT) to reconstitute haematopoietic Lrp1 expression. In this study, we have studied the metabolic phenotype of endothelial Lrp1 knockout mice at basal condition and during high-fat-diet (HFD)-induced obesity. Interestingly, mice with Lrp1 depletion in ECs demonstrate improved metabolic responses. Mechanistically, Lrp1 promotes peroxisome proliferator-activated receptor-γ (Pparγ) activity by acting as a co-activator. Our results suggest a new receptor-dependent regulatory mechanism for Ppars and the endothelium plays a critical role in maintaining lipid and glucose homeostasis.

## Results

**HFD-induced body weight gain.** First, we evaluated the role of endothelial Lrp1 in a HFD-induced obesity model. Lrp1[flox/flox] (Lrp1[f/f]) mice were bred with Tie2Cre[+] transgenic mice to generate Lrp1[f/f]; Tie2Cre[+] (Cre+) mice, in which the Lrp1 gene is specifically deleted in ECs and bone marrow-derived haematopoietic cells (Supplementary Fig. 1a,c,d)[18]. Both Cre+ and their littermate control Cre− male mice were fed either control chow (CC) or HFD for 16 weeks. We found that Lrp1-depleted Cre+ mice were much leaner than Cre− mice following HFD feeding, with ~25.1% less body weight gain (Fig. 1a, Supplementary Fig. 1e). This marked reduction in body weight gain of Cre+ mice was contributed partly by the lower masses of liver and adipose tissue (Fig. 1b, Supplementary Fig. 1f). In addition, the number of lipid droplets in the liver was much less and the size of adipocytes in epididymal fat was significantly smaller in Cre+ mice than in Cre− mice (Fig. 1c–f). Interestingly, we observed similar phenotypes even after BMT to reconstitute wild-type (Wt) haematopoietic cells in Cre+

(Cre+/BMT) mice (Fig. 1g–i, Supplementary Fig. 1b), suggesting that changes in white adipose tissue and liver were mainly resulted from EC-specific Lrp1 depletion.

**Lipid and adipokine profiles and physical activity.** Liver and adipose tissues play crucial roles in lipogenesis, lipid clearance and storage of excess energy to maintain energy homeostasis. Therefore, we compared the circulating profiles of lipids and adipokines in Cre+/BMT and Cre−/BMT mice. Surprisingly, even before HFD feeding when no obvious body weight difference was observed (Fig. 1g), plasma LDL-cholesterol level was significantly higher and HDL-cholesterol level was lower in Cre+/BMT mice than in Cre−/BMT control mice (Fig. 2a,b, Supplementary Fig. 2a). Given that Lrp1 is a receptor of ApoE and chylomicron remnant and required for HDL secretion in the liver[5,6,16], our data suggest that endothelial Lrp1 may also contribute to lipid clearance or reverse cholesterol transport process. Following HFD feeding, all LDL-cholesterol, HDL-cholesterol, triglyceride (TG) and total cholesterol levels were decreased in Cre+/BMT mice, compared to Cre−/BMT control mice (Fig. 2a–d, Supplementary Fig. 2a–d), suggesting that endothelial Lrp1 plays distinctive roles in lipid metabolism at the physiological condition and in response to hyperlipidaemia stress. LPL activity was increased in Cre+/BMT mice, compared to Cre−/BMT control mice before and after HFD feeding (Supplementary Fig. 2e). These elevated LPL activities are inversely correlated with the decreased serum TG levels (Fig. 2c, Supplementary Fig. 2c,d), suggesting that Lrp1 depletion in ECs results in increased lipolysis. In addition, decreased levels of adiponectin and leptin were observed even before HFD feeding in Cre+/BMT mice compared to Cre−/BMT mice (Fig. 2e,f), which was positively correlated to the decreased adipocyte hypertrophy (Fig. 1e,f). We also observed that Cre+/BMT mice consumed similar amount of food as Cre−/BMT control mice along the whole process of HFD feeding (Fig. 2g), suggesting that the appetite was not altered but leptin sensitivity might increase in response to endothelial Lrp1 depletion. Interestingly, Cre+/BMT mice were physically more active, displaying increases in oxygen consumption and locomotor activity, particularly during the night time (Fig. 2h–j, Supplementary Fig. 2f,g). This increased physical activity, which might lead to increased oxidative metabolism of TG-derived fatty acids in the heart and skeletal muscle, likely explains the increased lipolysis (Supplementary Fig. 2e) and decreased TG level in Cre+/BMT mice (Fig. 2c). Taken together, our data suggest that endothelial Lrp1 depletion regulates energy balance by shifting the metabolic cycle towards catabolism instead of energy storage.

**Insulin sensitivity and glucose tolerance.** Since elevated TG level is commonly observed in type 2 diabetes and obesity patients and insulin sensitivity is negatively correlated with TG level[19], lower TG level in Cre+/BMT mice (Fig. 2c) implicates that insulin sensitivity may be increased in Lrp1 knockout mice. Therefore, we examined how Cre+/BMT and Cre−/BMT mice responded to glucose and insulin challenges. There was no significant difference in fasting blood glucose concentration in Cre+/BMT and Cre−/BMT mice (Fig. 2k). However, the insulin level was 56.3% lower in Cre+/BMT mice after HFD feeding (Fig. 2l). In addition, Cre+/BMT mice displayed higher insulin sensitivity and a more efficient clearance of systemic glucose before or after HFD feeding (Fig. 2m–o, Supplementary Fig. 2h), suggesting that Lrp1 in endothelium plays a role in insulin homeostasis, likely indirectly through the regulation of weight gain or other mechanisms.

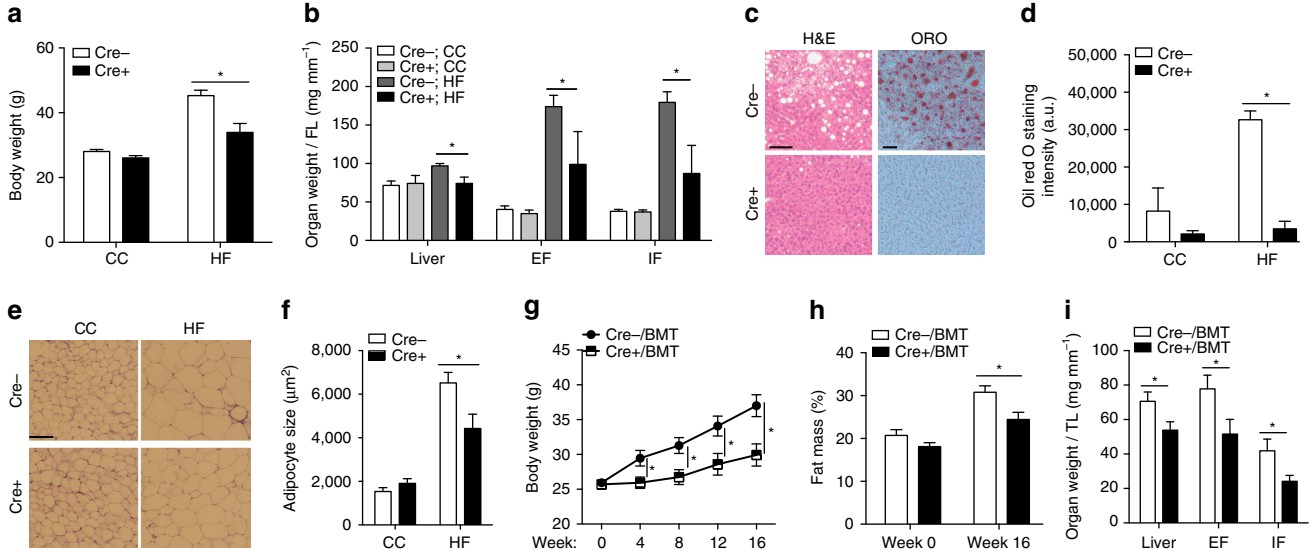

**Figure 1 | EC-specific Lrp1 knockout mice display less weight gain than their littermate control upon HFD feeding.** (**a**) Changes in the body weight of Lrp1$^{f/f}$;Tie2Cre$^{+/-}$ (Cre + or Cre − ; generated by the breeding of B6;129S7-Lrp1$^{tm2Her}$/J and B6.Cg-Tg(Tek-cre)1Ywa/J mice) mice were monitored following 16 weeks of CC or HFD feeding. (**b**) The liver, epididymal fat (EF) and inguinal fat (IF) tissue masses of Cre + mice were significantly lower following HFD feeding for 16 weeks, compared to Cre − mice. The tissue masses were normalized by femur length (FL). (**c-d**) The staining of liver with Oil Red O (ORO) demonstrates significantly decreased lipid deposition in Cre + mice following HFD feeding for 16 weeks. Images presented in **c** are representative results of four sections per mouse in each group. The quantitative analysis of ORO-stained signals is shown in **d**. (**e,f**) The H&E staining of epididymal fat indicates smaller adipocyte sizes for Cre + mice, compared to Cre − mice following HFD feeding for 16 weeks. Images presented in **e** are representative results of six sections per mouse in each group. The sizes of adipocytes were quantified and shown in **f**. (**g**) Mice with EC-specific Lrp1 depletion (mice with Tie2Cre-mediated deletion and BMT; Cre + /BMT) display lower body weight gain. Body weight of Cre − /BMT or Cre + /BMT mice was measured along the whole course of HF diet feeding. (**h**) Body composition of mice before and after 16-week HFD feeding was measured by the dual energy X-ray absorptiometry. (**i**) The masses of liver, epididymal fat (EF) and inguinal fat (IF) tissues were significantly lower in Cre + /BMT mice on HFD for 16 weeks, compared to Cre − /BMT mice. Tissue masses were normalized by tibia length (TL). $n = 4$ for Cre + mice and 5 for Cre − mice and $n = 8$ for Cre + /BMT mice and 7 for Cre − /BMT mice. *$P < 0.05$. Scale bars, 50 µm in **c** and 100 µm in **e**. Analysis was two-way analysis of variance followed by Bonferroni (for **a**) and Fisher's least significant difference (for **b,d,f,g,h**) multiple comparison test and unpaired Student's t-test (for **i**).

**Lrp1 is a co-activator of Pparγ.** Lrp1 is a heterodimer composed of a 515-kDa α chain (Lrp1α), which possesses four extracellular ligand-binding domains (LBDs), and an 85-kDa membrane-anchored cytoplasmic β chain (Lrp1β), which remains non-covalently associated with Lrp1α and regulates intracellular signalling by interacting with receptors or adaptor proteins in the cytoplasm[1]. Lrp1β can also be processed by γ-secretase and translocated to the nucleus, in which it interacts with and inhibits transcriptional regulators, such as interferon regulatory factor 3 (IRF-3) and poly(ADP-ribose) polymerase-1 (refs 11,20,21). Given that Pparγ is a master regulator of energy homeostasis and EC-specific deletion of Pparγ displays similar metabolic phenotypes to what we observed in Cre + /BMT mice[22], we tested whether Lrp1 might interact with Pparγ and regulate its transcriptional activity. Indeed, we demonstrated that the intracellular C-terminal domain (ICD) of Lrp1β bound to the LBD of Pparγ (Fig. 3a–c,i). More importantly, overexpression of Lrp1β significantly increased Pparγ transcriptional activity (Fig. 3d). To determine the regulatory role of Lrp1 in vivo, we bred Lrp1$^{f/f}$; CAG-CreER$^{+/-}$ mice with PPRE-luc$^+$ reporter mice to create Lrp1$^{f/f}$; CAG-CreER$^{+/-}$; PPRE-luc$^+$ (CAG-Cre + / − ;PPRE-luc + ) mice and measured the Ppar-responsive luciferase activity in response to tamoxifen-mediated Lrp1 depletion. Upon injection of the luciferase substrate D-luciferin, bioluminescent signal was detected in CAG-Cre − ; PPRE-luc + mice; however, the signal in Lrp1 depleted CAG-Cre + /PPRE-luc + mice was ∼3.76-fold lower than the CAG-Cre − ;PPRE-luc + control mice (Fig. 3e,f). In response to Pparγ agonist pioglitazone, bioluminescence intensity was further increased, indicating the activation of Pparγ in vivo.

Again, Lrp1 depletion significantly inhibited the luciferase activity in response to pioglitazone (Fig. 3e,f). This in vivo Pparγ activity assay demonstrates for the first time that Lrp1 is required for Pparγ transcriptional activity. Moreover, we discovered that Lrp1 also regulated the expression of Pparγ's target gene—pyruvate dehydrogenase kinase 4 (Pdk4; Fig. 3g). Using chromatin immunoprecipitation (ChIP) assays, we detected that Lrp1 was in a complex with the promoter of Pdk4 at basal condition and their association was increased in response to pioglitazone or palmitic acid treatments (Fig. 3h). Taken together, we conclude that Lrp1β interacts with Pparγ and positively regulates its transcriptional activity. Finally, we demonstrated that Lrp1β also bound to Pparα and Pparβ/δ, promoted their transcriptional activities and induced their target gene Pdk4 in vivo (Supplementary Fig. 3), suggesting that the promoting effect of Lrp1 is also applicable for other Ppars.

It is known that Ppar co-activators promote Ppar transcriptional activity by forming a nuclear receptor/co-activator complex[23]. A conserved LXXLL motif in these co-activators is required for their interaction with the LBD of nuclear receptors, such as Pparγ[24,25]. Interestingly, there is a VGGLL sequence located in a coil region of Lrp1β-ICD (Supplementary Fig. 4), which is very similar to the LXXLL motif in Pparγ co-activators, such as SRC-1 (Supplementary Table 1)[24,25]. To examine whether this VGGLL motif is required for its interaction with Pparγ, we generated a mutant Lrp1β (Lrp1β-Mut), in which the two conserved leucines were replaced by two alanines. Co-immunoprecipitation experiment indicated that Lrp1β-Mut pulled down much less glutathione S-transferase (GST)-tagged Pparγ-LBD than Lrp1β-Wt (Fig. 3i,j), suggesting that VXXLL

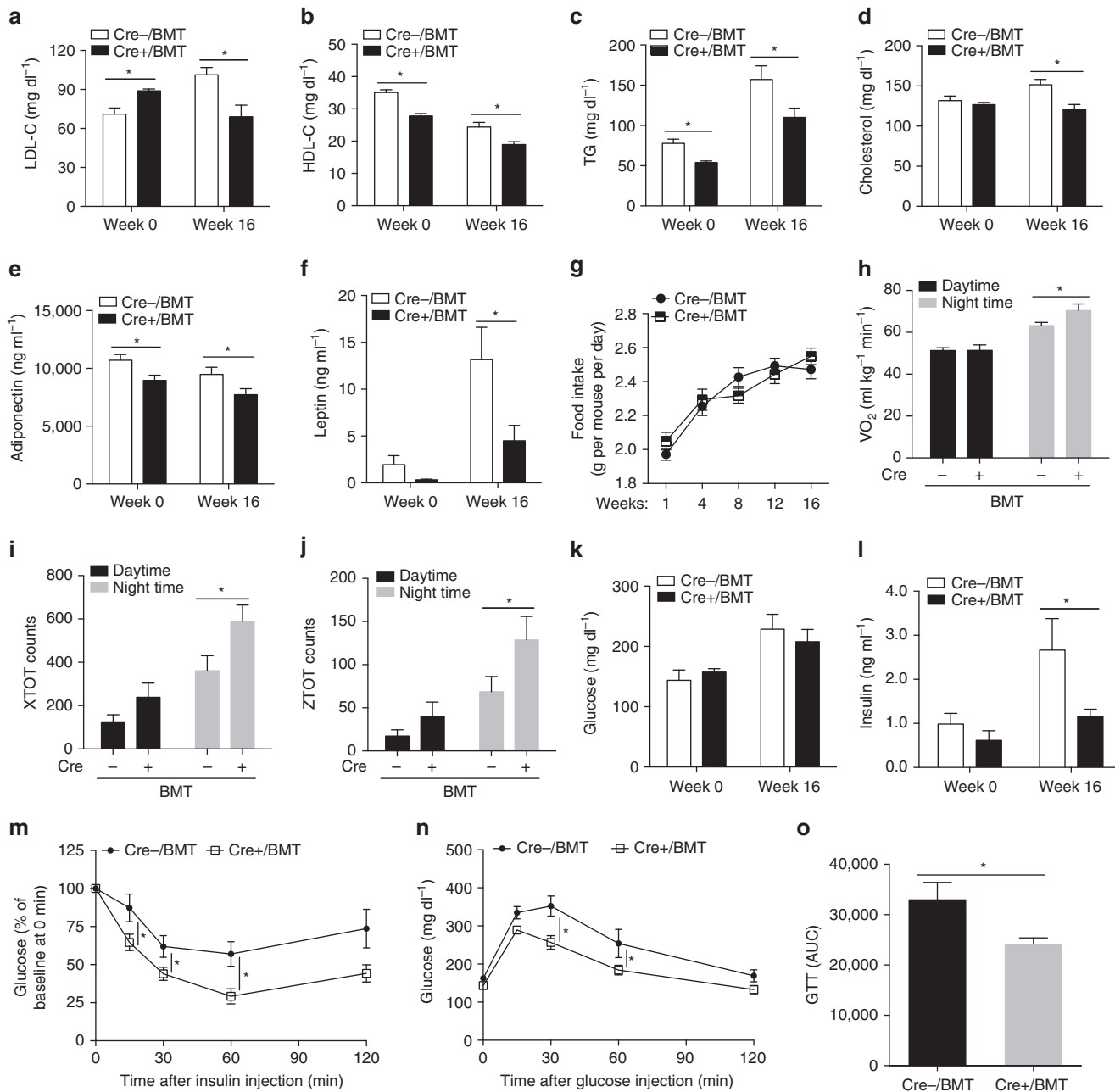

**Figure 2 | EC-specific Lrp1 depletion in mice results in improved metabolic responses.** (a–f) Plasma levels of LDL-cholesterol (LDL-C), HDL-cholesterol (HDL-C), TG, total cholesterol, adiponectin and leptin were analysed in Cre + and Cre − mice before and after HFD feeding for 16 weeks (Weeks 0 or 16, respectively). (g) Daily food intake was monitored for both Cre + /BMT and Cre − /BMT mice individually along the whole period of HFD feeding. (h–j) VO$_2$ (h), locomotor activity in x axis (XTOT, i) and z axis (ZTOT, j) were measured in mice by metabolic cage studies before HFD feeding. (k,l) Plasma levels of fasting glucose and insulin were analysed in Cre + /BMT and Cre − /BMT mice before and after HFD. (m) Insulin tolerance tests were performed with Cre + /BMT and Cre − /BMT mice before HFD feeding (Week 0). (n,o) Glucose tolerance tests (GTT) were performed with Cre + /BMT and Cre − /BMT mice after HFD feeding for 16 weeks (Week 16). The area under the curve for the glucose tolerance test was quantified and presented in o. $n = 8$ for Cre + /BMT mice and 6 for Cre − /BMT mice. *$P < 0.05$. Analysis was two-way analysis of variance followed by Fisher's least significant difference multiple comparison test (for a–n) and unpaired Student's t-test (for i).

motif is required for Lrp1β and Pparγ interaction. Furthermore, in contrast to Lrp1β-Wt, which significantly increased the transcriptional activity of Pparγ, Lrp1β-Mut dramatically inhibited it (Fig. 3k). The ChIP assay further demonstrated that the association of Lrp1-Mut with the Pdk4 promoter was significantly decreased, compared to that of Lrp1-Wt (Fig. 3l). Therefore, we conclude that Lrp1β acts as a transcriptional co-activator of Pparγ.

**Lrp1 is required for Pparγ activation in ECs.** Endothelial Pparγ regulates the expression of lipid-handling genes in ECs[22]. Since our data suggest that Lrp1β acts as a novel transcriptional co-activator of Pparγ, it is plausible that depletion of Lrp1 in ECs may also lead to dysregulation of lipid-handling genes. To examine this hypothesis, we first confirmed that Lrp1β interacted with endogenous Pparγ in ECs, and their interaction was increased mildly upon pioglitazone treatments for 5–15 min

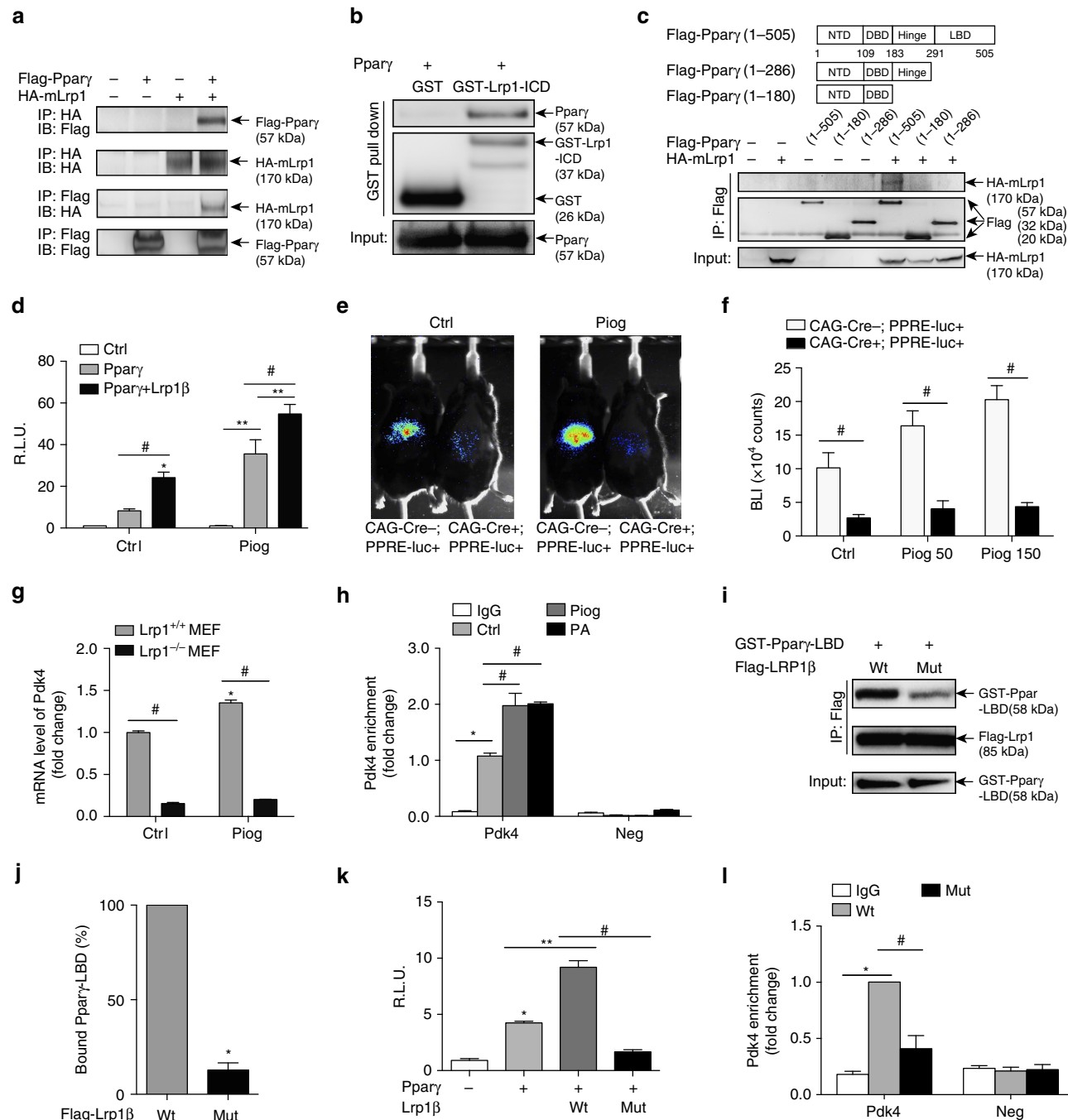

**Figure 3 | Lrp1β binds to Pparγ and promotes its transcriptional activity.** (**a**) Lysates of HEK 293 cells with overexpressed Flag-tagged Pparγ (Flag-Pparγ) and HA-tagged mini-Lrp1 receptor (HA-mLrp1[13]) were immunoprecipitated (IPed) and blotted with the indicated antibodies. (**b**) GST pull-down assay was performed with recombinant Pparγ protein. GST-Lrp1-ICD: GST-tagged Lrp1 ICD (GST-ICD; a.a. 4445–4544 of human Lrp1). (**c**) Co-IP assay was performed with lysates of HEK293 cells that were transfected with the indicated constructs. (**d**) Ppar reporter assay was performed with lysates of HEK293 cells that were treated with 10 μM pioglitazone (Piog). (**e,f**) In vivo PPRE-driven luciferase activity was evaluated with Lrp1[f/f];CAG-CreER[+/−];PPRE-luc[+] (CAG-Cre+;PPRE-luc+ and CAG-Cre−;PPRE-luc+ generated by the breeding of B6;129S7-Lrp1[tm2Her]/J, B6.Cg-Tg(CAG-cre/Esr1*)5Amc/J and repTOP PPRE-Luc mice) mice. Pioglitazone, 50 or 150 mg kg[−1]. Images presented in **e** are representative results of the indicated mouse group and bioluminescence intensity (BLI) was quantified in **f**. (**g**) Pdk4 mRNA level was measured with real-time PCR assays. Lrp1 knockout MEFs (Lrp1[−/−] MEFs) and Wt (Lrp1[+/+] MEFs) were treated with 10 μM pioglitazone. (**h**) ChIP-qPCR was performed for endogenous Pdk4 promoter and negative DNA region control (Neg). Stable Lrp1β-transfected HEK293 cells were treated with pioglitazone (Piog) at 10 μM or palmitic acids (PA) at 0.5 mM for 24 h. Enriched chromatin fractions were IPed with Lrp1 or IgG. (**i,j**) HEK293 cells were transfected with Flag-Lrp1β-Wt (VGGLL) or Mut (VGGAA). Their lysates were mixed with GST fused recombinant Pparγ LBD (Pparγ-LBD) protein for IP and immunoblotting. The bound Pparγ-LBD protein level was quantified by measuring its band intensity and normalized by the amount of enriched Flag-Lrp1β protein (**j**). (**k**) Ppar reporter assay was performed in HEK293 cells. (**l**) HEK293 cells were transfected with Flag-tagged Lrp1β-Wt or Mut and ChIP assays were performed with Flag antibody or IgG. n = 3 for cell culture experiments in **a–d,g–l**. n = 3 for Lrp1[f/f];CAG-CreER[+];PPRE-luc[+] mice and 4 for Lrp1[f/f];CAG-CreER[−];PPRE-luc[+] mice in **e,f**. *P < 0.05, compared to control cells. ** or #P < 0.05. Analysis was two-way analysis of variance (ANOVA) followed by Fisher's least significant difference multiple comparison test (for **d,f–h,l**), one-way ANOVA followed by Bonferroni (for **k**) and unpaired Student's t-test (for **j**).

(Fig. 4a, Supplementary Fig. 5a). We then examined the expression of Pparγ-dependent lipid-handling genes in mouse cardiac-derived ECs (MEC) and mouse primary ECs. As expected, knockdown or knockout of Lrp1 significantly inhibited Pparγ activity and decreased the expression of Pparγ target genes, such as Cd36, Pdk4 and CCAAT/enhancer binding protein alpha (C/ebpα; Fig. 4b–f, Supplementary Fig. 5b,c). More importantly, following tamoxifen treatment, isolated Lrp1-depleted ECs from CAG-Cre + ;PPRE-luc + reporter mice demonstrated significantly lower Pparγ activity than CAG-Cre − ;PPRE-luc + ECs (Fig. 4c). Excitingly, the decreased levels of Cd36, Pdk4 and C/ebpα could be 'rescued' by overexpressed Lrp1β-Wt but not by Lrp1β-Mut (Fig. 4e,f), suggesting that the VXXLL motif of Lrp1 is required for Pparγ-dependent gene induction.

We also tested whether Lrp1 regulates Pparγ target genes expression by knocking down Pparγ-treated cells with Pparγ agonists. As expected, treatments of overexpressed Lrp1β, pioglitazone or both increased mRNA levels of Pparγ target genes. However, these increases were all inhibited in Pparγ knockdown ECs (Fig. 5a). On the other hand, Pparγ agonists including different thiazolidinediones (TZDs; pioglitazone, ciglitazone, rosiglitazone and troglitazone) and palmitic acids increased the mRNA levels of Pparγ target genes Cd36, Pdk4 and C/ebpα. However, these increases were inhibited in Lrp1-depleted ECs (Fig. 5b). We also isolated ECs from endothelial Lrp1-depleted mice following HFD feeding for 9 weeks. Consistently, mRNA levels of Cd36, Pdk4 and C/ebpα were increased in Cre − control ECs in response to HFD feeding. However, these increases were abolished in Cre + cells (Fig. 5c). In addition to these in vitro experiments, we investigated whether Pparγ agonists affect metabolic phenotypes of endothelial Lrp1-depleted mice. Specifically, we analysed metabolic parameters, energy expenditure, insulin and glucose tolerance responses in endothelial Lrp1 knockout or their littermate control mice following the treatment of pioglitazone or rosiglitazone for 4 and 3 weeks, respectively. Our results demonstrate that, in response to pioglitazone and rosiglitazone, most of metabolic phenotypes resulted from endothelial Lrp1 depletion were still detected in these mice, compared to Cre − /BMT mice (Fig. 6, Supplementary Fig. 6). Taken all together, our additional data provide further mechanistic support for the hypothesis that Lrp1 is a co-activator of Pparγ and required for its ligand-dependent target gene induction.

**Pparγ/Cd36 mediated cholesterol internalization.** Cd36 is a key scavenger receptor that is required for internalization of oxidized LDL (oxLDL) in ECs[26]. Consistently, we demonstrated that Lrp1 and its VXXLL motif were required for cholesterol internalization in ECs upon oxLDL loading (Fig. 7a–c). We also performed oxLDL loading assays with Pparγ knockdown ECs. Our results indicated that Pparγ was required for cholesterol internalization induced by overexpression of Lrp1, pioglitazone treatment or both (Fig. 7d). In response to Pparγ agonists pioglitazone and palmitic acids, cholesterol uptake was increased. However, these increases were blocked in Lrp1-depleted ECs (Fig. 7b,e).

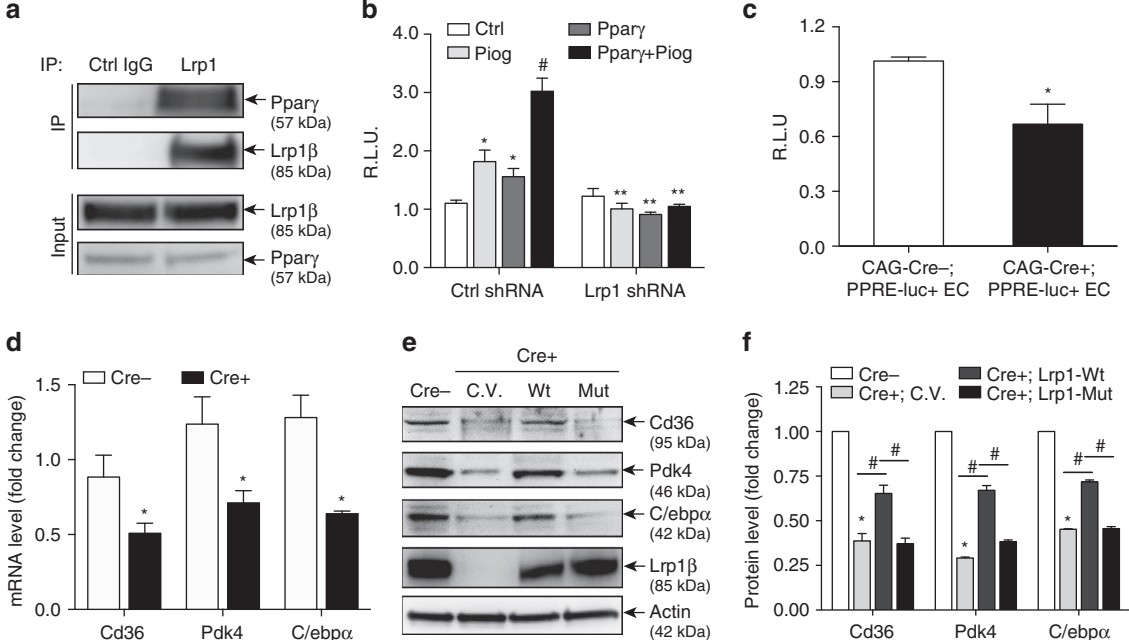

**Figure 4 | Lrp1β regulates Pparγ-dependent lipid and glucose gene induction in ECs.** (**a**) Lrp1β interacts with Pparγ in ECs. Lysates of mouse cardiac-derived ECs (MECs) were immunoprecipitated with anti-Lrp1 C-terminal antibody or control IgG and analysed by western blotting. (**b**) Lrp1 knockdown in MECs blocks Ppar transcriptional activity. The PPRE-Luc, renilla and Flag-Pparγ constructs were transfected into Lrp1 shRNA or control shRNA stably transfected MECs and then cells were treated with 10 μM pioglitazone. n = 3. *P < 0.05, compared to control cells. #P < 0.05, compared to control sh-MECs transfected with Flag-Pparγ or treated with pioglitazone. **P < 0.05, compared to control sh-MECs with same treatment or transfection. (**c**) The luciferase enzymatic activity of mouse ECs isolated from Lrp1f/f;CAG-CreER − ;PPRE-luc + (CAG-Cre − ;PPRE-luc + EC) and Lrp1f/f;CAG-CreER + ; PPRE-luc + (CAG-Cre + ;PPRE-luc + EC) mice was measured. (**d**) The mRNA levels of Cd36, Pdk4 and C/ebpα in primary mouse ECs isolated from Lrp1f/f;CAG-CreER + (Cre + ) or Lrp1f/f;CAG-CreER − (Cre − ) mice were measured with PCR assays. Same cells were used in **e,f**. (**e,f**) Protein levels of Cd36, Pdk4 and C/ebpα were measured with western blotting. Before that, isolated primary mouse ECs were transfected with control vector (C.V.) or the indicated plasmids. The band intensity of proteins was normalized to actin (**f**). In **a**,**c–f**, n = 3. *P < 0.05, compared to Cre − cells. #P < 0.05. Analysis was two-way analysis of variance followed by Fisher's least significant difference multiple comparison test (for **b,f**) and unpaired Student's t-test (for **c,d**).

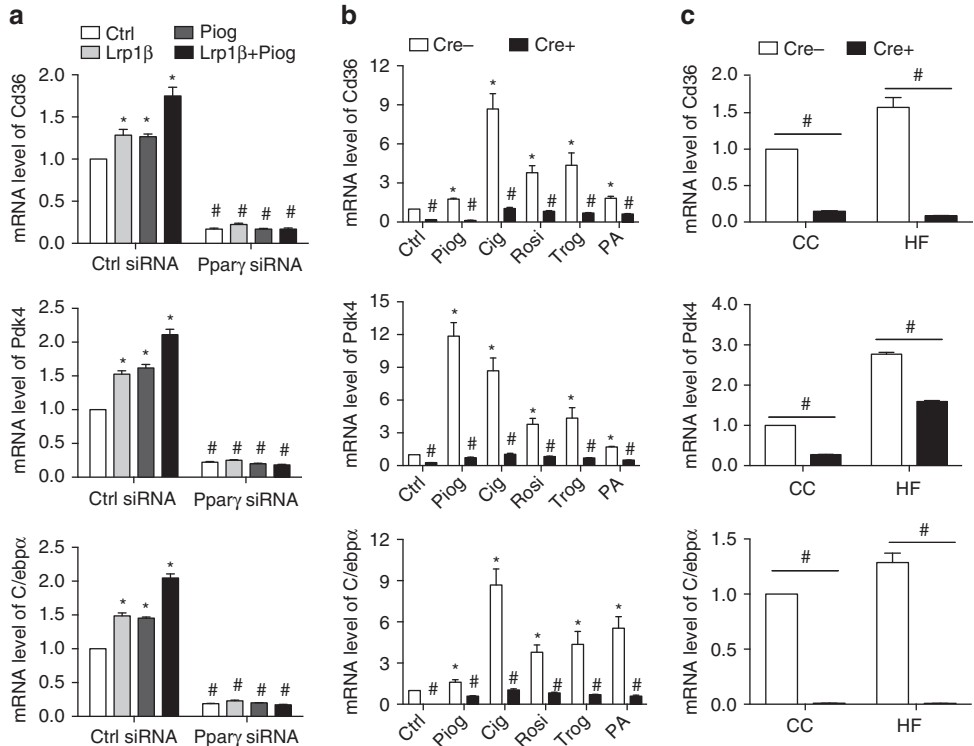

**Figure 5 | mRNA levels of Pparγ target genes are regulated by Lrp1 depletion in ECs.** The mRNA levels of Cd36, Pdk4 and C/ebpα in primary mouse ECs were measured with real-time PCR assays. Before PCRs, in **a**, cells were transfected with Pparγ-specific or control siRNAs and also transfected with Lrp1β or treated with pioglitazone at 10 μM or both. In **b**, Cre + or Cre − ECs were treated with pioglitazone at 10 μM, ciglitazone at 20 μM, rosiglitazone at 20 μM, troglitazone at 50 μM and palmitic acids at 0.5 mM. In **c**, cells were isolated from Cre + or Cre − mice after HFD feeding for 9 weeks. $n = 3$ for primary ECs transfected with either control or Pparγ siRNAs in **a** and Cre + or Cre − ECs in **b**. $n = 4$ for Cre + mice and 3 for Cre − control mice in **c**. *$P < 0.05$, compared to control siRNA-transfected cells or Cre − control cells. #$P < 0.05$, compared to control siRNA-transfected cells or Cre − ECs upon same treatments. Analysis was two-way analysis of variance followed by Bonferroni's multiple comparison test (for **a,c**) and multiple unpaired Student's $t$-test followed by Holm–Sidak correction (for **b**).

This suggests that Pparγ activity mediates Lrp1-dependent cholesterol internalization. Last, we tested cholesterol uptake with ECs isolated from 9-week high-fat-fed mice. We discovered that cholesterol uptake was significantly decreased in Lrp1 knockout ECs in CC-fed mice, compared to Cre − control cells (Fig. 7f). However, very surprisingly, cholesterol uptake was increased dramatically in Lrp1 knockout ECs isolated from HFD-fed mice, compared to Cre − control ECs isolated from HFD-fed mice or Lrp1 knockout ECs isolated from CC-fed mice (Fig. 7f). This increase, inversely correlated to decreased LDL-cholesterol level following HFD feeding in endothelial Lrp1 knockout mice (Fig. 2a, Supplementary Fig. 2a,b), could not be explained by the decreased induction of Cd36 in the same cells (Fig. 5c). It suggests that endothelial Lrp1 plays an active role in LDL-cholesterol clearance at basal condition and in response to hyperlipidaemia. Cd36 is required for Lrp1-mediated cholesterol internalization at basal condition. However, hyperlipidaemia stress activates Cd36-independent mediators for cholesterol internalization. This, together with our other observations such as the potential roles of endothelial Lrp1 in HDL and TG homeostasis, will become our future research directions. Nevertheless, our current results strongly suggest that endothelial Lrp1 plays a pivotal role in the regulation of metabolic homeostasis, at least partially, through the regulation of Pparγ transcriptional activation.

### Discussion

In this study, we present data supporting a new mechanistic model in which Lrp1 acts as a transcriptional co-activator of

Pparγ, and likely of all Ppars, in ECs. Also, this Lrp1β-dependent Pparγ activation establishes a missing link between Lrp1 and Pparγ signalling, which enables the 'outside-in' signalling in an efficient manner. It provides a new receptor-dependent regulatory mechanism for Ppars and suggests a direct involvement of gene transcription for a membrane receptor. It also supports that the endothelium plays a critical role in maintaining lipid and glucose homeostasis, expanding our present knowledge about what endothelial function and its dysregulation encompass. Depletion of Lrp1 in ECs results in similar phenotypes to its depletion in hepatocytes and adipocytes[5–7], suggesting a regulatory role of ECs in the liver and adipose tissue-mediated metabolic responses. Besides Cd36, other Lrp1/Pparγ target genes Pdk4 and C/ebpα are also known as critical regulators of lipid and glucose metabolism. The deficiency of Pdk4 lowers blood glucose and improves glucose tolerance and insulin resistance in mice[27,28]. C/ebpα is involved in adipogenesis and hepatic glucose and lipid metabolism[29,30]. However, their roles in ECs are largely unknown. It remains to be further characterized whether they play a role in the crosstalk between ECs and neighbouring cells.

Multiple changes of metabolic parameters have been observed with this EC-specific Lrp1 knockout (Cre + /BMT) mouse model, including changes of lipid profiles, such as LDL, HDL and TG. These changes happened even before the onset of HFD-induced obesity, suggesting that they might be directly resulted from Lrp1 depletion in the endothelium. Two hepatic Lrp1-deficient mouse models have been reported recently, one was generated by crossing Lrp1$^{flox/flox}$ mice with MX1-Cre transgenic mice and the deletion of Lrp1 was induced by polyinosinic acid-polycytidylic

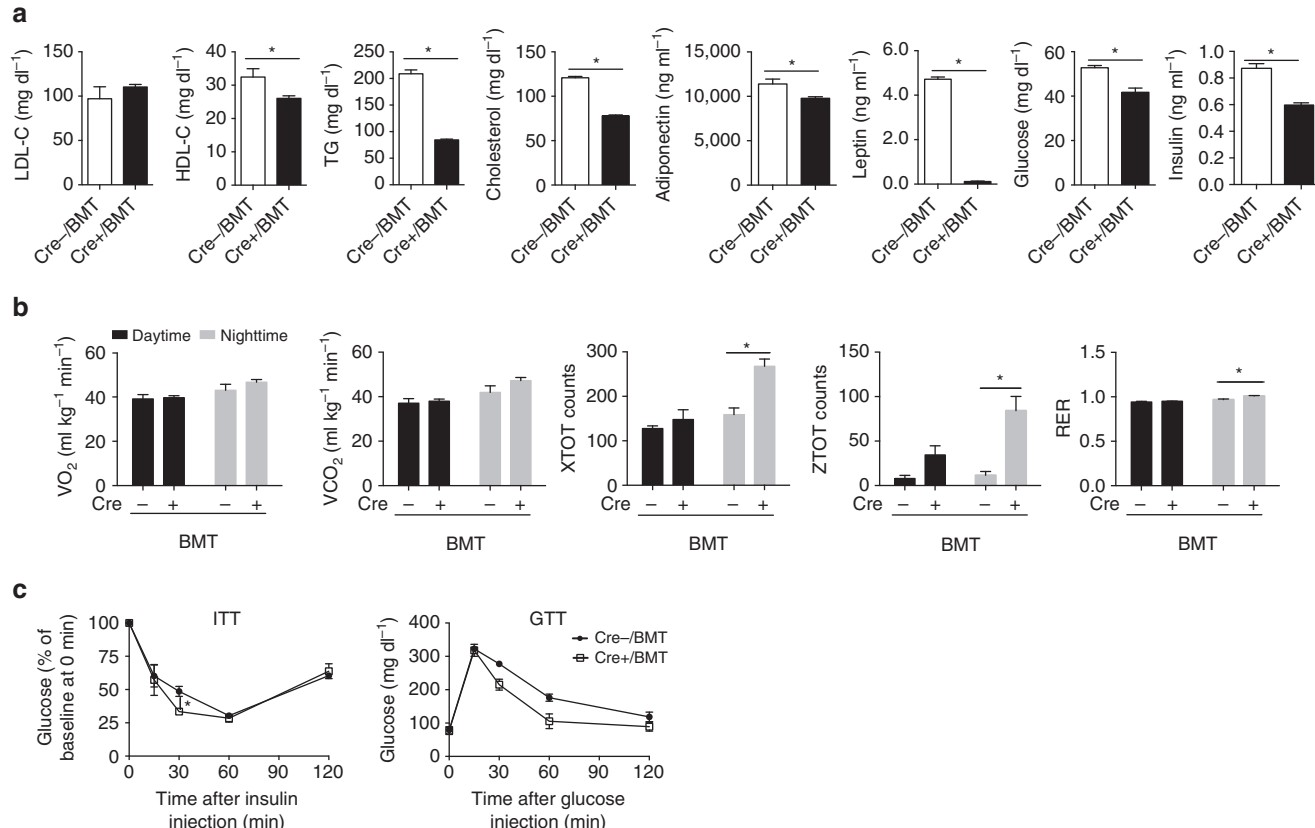

**Figure 6 | Mice with endothelial Lrp1 depletion display improved metabolic responses following the treatment of pioglitazone. (a)** Metabolic parameters including plasma levels of LDL-cholesterol (LDL-C), HDL-cholesterol (HDL-C), TG, total cholesterol, adiponectin, leptin, fasting glucose and insulin were analysed in Cre + /BMT and Cre − /BMT mice after pioglitazone treatments for 4 weeks. **(b)** $VO_2$, $VCO_2$, locomotor activity in x axis (XTOT) and z axis (ZTOT) and RER (respiratory exchange rate) were measured in mice by metabolic cage studies after the treatment of pioglitazone. **(c)** Insulin tolerance tests (ITT) and glucose tolerance tests (GTT) were performed with Cre + /BMT and Cre − /BMT mice after the treatment of pioglitazone. *$P < 0.05$. $n = 3$ for both Cre + /BMT mice and Cre − /BMT mice. Analysis was unpaired Student's t-test (for **a**) and two-way analysis of variance followed by Fisher's least significant difference multiple comparison test (for **b,c**).

acid injection[5]. The other was generated by crossing Lrp1[flox/flox] mice with albumin promoter-driven Cre (Alb-Cre) mice[6]. Given the fact that MX1-Cre also possesses recombinase activity in ECs[31], the phenotype observed in Lrp1[flox/flox]; MX1-Cre mice is due to the depletion of Lrp1 not only in hepatocytes but also in ECs. Interestingly, it was reported that LDL was increased in Lrp1[flox/flox]; MX1-Cre mice[5]. Similar phenotype was also observed in our endothelial Lrp1 knockout mice. However, it was not changed in Alb-Cre − mediated Lrp1 knockout mice[6], suggesting that endothelial Lrp1 may play a pivotal role in the regulation of LDL homeostasis. On the other hand, HDL levels were decreased in all three mouse models. It is well documented that hepatic Lrp1 regulates HDL production and plasma levels[6]; whether and how endothelial Lrp1 regulates HDL homeostasis remains to be further elucidated.

We have demonstrated that Lrp1 is a transcriptional co-activator of Pparγ. Therefore, it is not surprising that some aspects of the metabolic phenotype resulted from Lrp1 deletion in ECs are very similar to those with endothelial Pparγ deletion[22]. First, the induction of Pparγ's target gene Cd36 decreased in Lrp1 or Pparγ knockout ECs. Second, depletion of either of them in ECs resulted in decreased white adipose tissue mass, adipocyte size and improvements of insulin and glucose tolerance responses following HFD-induced obesity. Interestingly, specific deletion of Lrp1 or Pparγ in other tissues also results in similar phenotypes[7,32]. For example, mice with depletion of Lrp1 or Pparγ in adipocyte displayed lower fat mass and higher TG level.

Upon HFD feeding, both mice displayed lower body weight, decreased levels of adipokines, higher levels of food intake and energy consumption and improved insulin and glucose tolerance responses. Nevertheless, differences are also observed between Lrp1 and Pparγ tissue-specific knockout mouse models. For instance, endothelial Lrp1 knockout mice, but not Pparγ knockout mice, displayed decreased body weight gain (Fig. 1g)[22]. These differences are likely due to the differential pathways regulated by Lrp1 or Pparγ. Lrp1 has been reported to be a multifunctional receptor for many different ligands including ApoE and also acts as a co-receptor of signalling pathways, such as leptin signalling[8,16]. The divergence between Lrp1 and Pparγ signalling, although increasing the complexity for their in vivo functions, can be further dissected by studying how mice with Lrp1 depletion in the liver or fat respond to Pparγ agonists.

This study is, to our knowledge, the first to report that Lrp1 is a co-activator of Pparγ and that depletion of Lrp1 in endothelium regulates global energy homeostasis and alleviates some metabolic syndromes, such as obesity and insulin resistance. Pparγ agonists (TZDs) are popular drugs for the treatment of diabetes. However, substantial clinical and preclinical experiences have indicated that TZDs lead to a number of common adverse effects, including weight gain, fluid retention, congestive heart failure and bone fractures[33]. These side effects might be due to Pparγ-independent actions of TZDs[34]. Our studies may provide a more specific and safer therapeutic strategy by targeting endothelial Lrp1 and blocking its interaction with Pparγ.

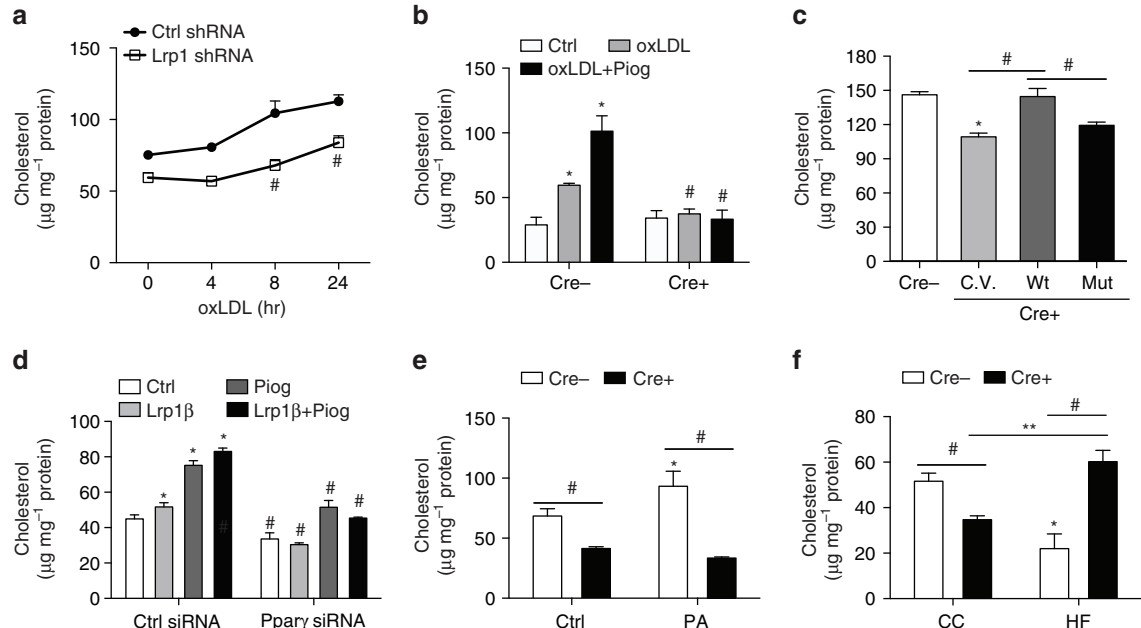

**Figure 7 | Lrp1β regulates Pparγ-dependent cholesterol internalization in ECs.** ECs following different treatments were treated with 20 μg ml$^{-1}$ oxLDL for determining internalized cholesterol contents. (**a**) Lrp1 shRNA or control shRNA stably transfected MECs were loaded with oxLDL for the indicated time periods. (**b**) Both isolated ECs were treated with oxLDL and 10 μM pioglitazone for 24 h. (**c**) Isolated primary MECs were transfected with the indicated plasmids. (**d**) Primary MECs were transfected with Pparγ-specific or control siRNAs and also transfected with Lrp1β or treated with pioglitazone at 10 μM or both. (**e**) Cre + or Cre − ECs were treated with palmitic acids at 0.5 mM for 24 h. (**f**) ECs were isolated from Cre + or Cre − mice after HFD feeding for 9 weeks. $n = 3$. *$P < 0.05$, compared to control shRNA or siRNA-transfected ECs or Cre − control cells. #$P < 0.05$, compared to control shRNA transfected ECs or Cre − cells upon same treatments, except indicated comparisons in **c**,**e**,**f**. **$P < 0.05$. Analysis was two-way analysis of variance (ANOVA) followed by Fisher's least significant difference multiple comparison test (for **a**,**b**,**d**–**f**) and one-way ANOVA followed by Bonferroni test (for **c**).

## Methods

**Mice.** B6;129S7-*Lrp1$^{tm2Her}$*/J (Lrp1-floxed; Lrp1$^{f/f}$) mice, B6.Cg-Tg(Tek-cre)1Ywa/J (Tie2Cre$^+$) and B6.Cg-Tg(CAG-cre/Esr1*)5Amc/J (CAG-CreER$^+$) mice were obtained from Jackson Laboratories (Bar Harbor, ME). The repTOP PPRE-Luc (PPRE-luc +) mice were kindly provided by Charles River Laboratories (Raleigh, NC). All mice were housed on a 12-h light/dark cycle, with food and water *ad libitum*. All experimental procedures on mice were performed according to the National Institutes of Health Guide for the Care and Use of Laboratory Animals and approved by the Institutional Committee for the Use of Animals in Research at Baylor College of Medicine. We used the mating of Lrp1$^{f/f}$ and Tie2Cre$^+$ mice to generate the Lrp1$^{f/f}$;Tie2Cre$^{+/-}$ (Cre + or Cre −) male mice for the HFD feeding studies. All male mice at 6–8 weeks old were fed the CC (14.7% calories from fat; PicoLab Rodent 50 no.5V5R, Lab Supply, Fort Worth, TX) or HFD (HF, 60% calories from fat; D12492, Research Diets Inc., New Brunswick, NJ) for 16 weeks. Pioglitazone or rosiglitazone injections were performed by intraperitoneal (i.p.) injections with a dosage of 20 mg kg$^{-1}$ day$^{-1}$ (pioglitazone) and 10 mg kg$^{-1}$ day$^{-1}$ (rosiglitazone) for 4 or 3 weeks, respectively, before measuring metabolic parameters. Body weight was monitored before and after they were fed with different diets. Blood serum was obtained before and after they were fed with different diets. We housed mice in single per cage for metabolic cage experiments and body weight measurements. We used the mating of Lrp1$^{f/f}$, CAG-CreER$^{+/-}$ and PPRE-Luc$^{+/-}$ mice to generate Lrp1$^{f/f}$;CAG-CreER$^{+/-}$;PPRE-luc$^{+/-}$ (CAG-Cre +/− ;PPRE-Luc +/− ) mice for *in vivo* bioluminescence imaging, primary microvascular EC isolation from the lung and heart and *in vitro* luciferase activity experiments.

**Body composition analysis.** We performed bone, lean and fat mass analysis with a GE Lunar PIXImus Body Composition Densitometer (GE Medical System).

**Bone marrow transplantation.** Bone marrow of Cre − control mice was transplanted into either Cre − or Cre + mice to generate mice either having or lacking Lrp1 expression in ECs only (Cre −/BMT or Cre +/BMT). The male Cre + and Cre − recipient mice at 6 weeks of age received a single semilethal dose of 900 rad irradiation using an RS2000 irradiator (Rad Source Technologies, Suwanee, GA). BM cells were harvested from the Cre − non-irradiated donor mice and $6 \times 10^6$ cells were injected via the tail vein into recipient mice 2 h after irradiation. The irradiated Cre + and Cre − mice without BMT died within 1 week, confirming the loss of haematopoietic cells. One month after BMT, both Cre + /BMT and Cre − /BMT mice were given HFD for 16 weeks to develop obesity and a series of metabolic parameters were then measured.

**Indirect calorimetry.** Mice were individually housed in metabolic chambers maintained at 20–22 °C on a 12-h light/dark cycle with lights on at 0700 hours. Metabolic measurements (oxygen consumption, food intake, locomotor activity) were obtained continuously using an Oxymax/CLAMS (Columbus Instruments) open-circuit indirect calorimetry system. Mice were provided with the CC or HFD and tap water *ad libitum*. Metabolic data were collected for 3–5 days following adaptation.

**Liver and adipose tissue histology.** The liver tissue samples were embedded in OCT (Tissue-Tek, Fisher Scientific, Pittsburgh, PA) and frozen on dry ice. Sections were stained with eosin and Oil Red O. For the immunofluorescence, the frozen sections of liver tissue were blocked with 5% heat-inactivated goat serum for 1 h, following by the overnight incubation with primary antibodies against CD31 (1:200 dilution; Cat. No. 553369 from BD Biosciences, San Jose, CA) and Lrp1 (1:200 dilution; 8G1, Cat. No. 20384 from Abcam, Cambridge, MA) diluted in the blocking solution. After three washes in Tris-buffered saline, cells were incubated in the dark with a second antibody conjugated with Alexa Fluor 488 or 568 (1:1,000 dilution; Molecular Probes, Eugene, OR) in blocking solution for 90 min at 37 °C. After three washes in Tris-buffered saline, the fragments were counterstained with 4,6-diamidino-2-phenylindole. The images were visualized by confocal laser scanning microscopy (Zeiss 780). For histology of adipose tissues, the paraffin sections was deparaffinized and rehydrated before being subjected to antigen retrieval. The adipose samples were soaked in boiling citric acid buffer (10 mmol l$^{-1}$, pH 6.0) for 9 min twice to expose the antigens. The immunofluorescence protocol follows the same procedure as that for liver tissue. The antibody against CD31 is from Abcam (1:100 dilution; Cat. No. 28364).

**Analysis of endocrine hormones and metabolites.** Mice were fasted overnight before blood sampling. Around 200 μl of blood was collected through sub-mandibular bleeding using a lancet. Plasma values for glucose were measured with an endocrine multiplex assay (Thermo Scientific, Waltham, MA) and insulin, leptin and adiponectin with ELISA kits (Millipore, Billerica, MA). The lipid contents were measured with Infinity TG and cholesterol kits (Thermo Scientific),

HDL/LDL cholesterol kits (Teco Diagnostics, Anaheim, California) and fast protein liquid chromatography (FPLC) was performed by the Mouse Metabolism Core Facility (Baylor College of Medicine).

**Lipoprotein lipase (LPL) analysis.** Serum samples were used to determine LPL activity using a fluorometric LPL Activity Assay Kit (Cell Biolabs, San Diego, CA, USA) according to the manufacturer's protocols.

**Glucose and insulin tolerance tests.** Glucose tolerance tests were performed after an overnight fasting. Blood glucose was measured before and 15, 30, 60 and 120 min after an i.p. glucose injection ($1\,g\,kg^{-1}$) with a Freestyle Glucose Monitoring System (Abbott Laboratories). Insulin tolerance testing was performed after a 6-h fast. Blood glucose was measured before and 15, 30, 60 and 120 min after an i.p. insulin injection ($0.75\,U\,kg^{-1}$; Novolin R, Novo Nordisk Inc.).

**Bioluminescence reporter imaging.** Lrp1^f/f^;CAG-CreER^+/−^;PPRE-luc^+^ (CAG-Cre^+/−^;PPRE-luc^+^) mice were used for this in vivo bioluminescence reporter imaging following previous protocol[35]. To initiate the study, we deleted Lrp1 by infusing mice i.p. with tamoxifen ($20\,mg\,kg^{-1}\,day^{-1}$, in corn oil, Sigma) for 5 consecutive days. One week later, mice were visualized with a Bruker In-Vivo MS FX Pro imager. For the detection of bioluminescence, mice were anaesthetized using infused isoflurane. The mice then received an i.p. injection of $50\,mg\,kg^{-1}$ D-luciferin (Cayman Chemical, Ann Arbor, Michigan). For pioglitazone treatment, mice received an i.p. injection of 50 or $150\,mg\,kg^{-1}$ pioglitazone (Sigma) 6 h before the imaging. Mice were placed in the light tight chamber under anaesthesia with infused isoflurane and a grey scale photo of the animals was first taken with dimmed light. The images were acquired immediately and every 5 min for 20 min. The bioluminescence intensity was detected with peaking around 5 min after D-luciferin injection and started to decay. The highest bioluminescence intensity counts were used for quantification.

**Reagents.** Lrp1 (C-terminal) antibody against a.a. 4532–4544 of human LRP1 was obtained from Sigma (Lrp1-CTD, Cat. No. L2170; St Louis, MO) and used for western blotting and immunoprecipitation. Another Lrp1 (1:200 dilution; 8G1, Cat. No. ab20384) antibody against a.a. 1–172 was purchased from Abcam and used for immunostaining. Lrp1 minireceptor construct (mLrp1) was generated as described[36] and generously provided by Dr Bu (Mayo Clinic, Jacksonville, FL). Flag-tagged Lrp1β was cloned into the pCMVTag2 vector (Life Technologies, Grand Island, NY). The mutation construct of Lrp1β (Lrp1-Mut) was generated by the QuikChange Multi-Site Directed Mutagenesis Kit (Agilent, Santa Clara, CA), where VXXLL motif located at a.a. 4484–4488 of human LRP1 protein was mutated to VXXAA. The mutation was confirmed by DNA sequencing. Flag-tagged Pparγ was cloned into the pCMV-Tag2 vector (Life Technologies). GST-tagged Lrp1 ICD (GST-ICD; a.a. 4445–4544 of human LRP1) construct was generated by subcloning of the fragment into pGEX-KG vector (ATCC, Manassas, VA). Antibodies for Cd36 (1:200 dilution; Cat. No. sc-9154) and Pdk4 (1:200 dilution; Cat. No. sc-130841) were purchased from Santa Cruz Biotechnology (Santa Cruz, CA). The C/ebpα antibody was obtained from Cell Signaling Technology (1:1,000 dilution; Cat. No. 2295, Denvers, MA). Pioglitazone, ciglitazone, rosiglitazone, troglitazone and palmitic acid were purchased from Sigma.

**Cell culture and isolation of primary ECs.** HEK293 cells (Cat. No. CRL-1573, ATCC) and MECs[13] were grown in DMEM supplemented with 10% foetal bovine serum (FBS) and antibiotics ($100\,U\,ml^{-1}$ penicillin, $68.6\,mol\,l^{-1}$ streptomycin). Mouse primary microvascular CAG-Cre^+/−^;PPRE-luc^+^ or Cre^+/−^ ECs were isolated from Lrp1^f/f^;CAG-CreER^+/−^;PPREluc^+^ or Lrp1^f/f^;CAG-CreER^+/−^ mice at 1–2 weeks old, using PECAM-1 antibody (BD Biosciences) Dynabead selection as described before[37]. ECs were cultured in MCDB131 medium supplemented with growth factors, hydrocortisone, 10% FBS and antibiotics ($100\,U\,ml^{-1}$ penicillin, $68.6\,mol\,l^{-1}$ streptomycin). Isolated ECs were further confirmed by CD31 staining and pathogen free. Lrp1 knockout mouse embryonic fibroblasts (MEFs; PEA 13, CRL-2216) and control MEFs (MEF-1, CRL-2214) were purchased from ATCC and cultured in DMEM supplemented with 10% FBS and antibiotics ($100\,U\,ml^{-1}$ penicillin, $68.6\,mol\,l^{-1}$ streptomycin). The depletion of Lrp1 protein was induced by treating cultured cells with 4-hydroxytamoxifen at $1\,\mu M$ for 5 days. For the construction of stable Flag-tagged Lrp1β and control vector-containing HEK293 cell lines, 50–70% confluent HEK293 cells in six-well plates were transfected with Flag-tagged Lrp1β with Lipofectamine 2000 (Life Technologies) and then positive cells were selected by using Geneticin (G418; Life Technologies). The generation of stable MEC cell lines was described in our previous publication[13].

**Immunoblotting and immunoprecipitation.** Cells were harvested in lysis buffer (1% Triton X-100, 50 mM Tris, pH 7.4, 150 mM NaCl, 1 mM Na$_3$VO4 and 0.1% protease inhibitor mixture; Sigma) and clarified by centrifugation at 16,000g. Equal amounts of protein were incubated with a specific antibody overnight at 4 °C with gentle rotation. Protein A/G Plus-agarose beads (Santa Cruz Biotechnology) were used to pull down the antibody complexes following previously described methods[11]. Proteins were separated by SDS–polyacrylamide gel electrophoresis and transferred to nitrocellulose membranes. Uncropped western blotting images for figures are shown as Supplementary Fig. 7.

**Luciferase assay.** HEK293 cells or MECs were transfected with PPRE-responsive firefly luciferase and constitutively expressing renilla plasmids. One day later, cells were treated with the indicated reagents. Cells were lysed 24 h later and analysed with the Dual-Luciferase Reporter Assay System from Promega (Madison, WI) according to the manufacturer's instructions using a Tecan Infinite 200 Pro microplate reader.

**Chromatin immunoprecipitation.** Cells were crosslinked with 1% formaldehyde and then sonicated with a Vibra-Cell sonicator (Sonics & Materials Inc., Newtown, CT). Soluble nuclear material from approximately million cells was used per immunoprecipitation. ChIP was performed using Magna ChIP Protein A + G magnetic beads (Millipore, Billerica, MA) and precipitated with Lrp1 antibody, Flag antibody or IgG as the control. Eluted DNA was isolated using the PCR Purification Kit (Qiagen). PCR was performed on DNA samples using the Quantifast SYBR Green PCR Kit (Qiagen) and validated primers amplifying the promoter region of Pdk4 (forward primer: 5′-ccacgttgccccagatacct-3′ and reverse primer: 5′-cactggaacttggaaacgcgt-3′) and the negative DNA region control in the vicinity of the Pdk4 promoter (forward primer: 5′-ggctcttttcgttccctctc-3′ and reverse primer: 5′-cttcaaagacgggagacag-3′) in Roche Lightcycler 480 PCR machine. Reaction mixtures were incubated at 95 °C for 5 min followed by 45 cycles at 95 °C for 10 s, 50 °C for 10 s and 72 °C for 10 s. Results were expressed relative to input control and normalized to control samples.

**Real-time PCR.** The RNA was reverse-transcribed into cDNAs with the iScript cDNA Synthesis Kit (Bio-Rad, Hercules, CA, USA). The specific pairs of primers used for the real-time PCR are the following: Lrp1 (forward primer: 5′-ggaccacc atcgtggaaa-3′ and reverse primer: 5′-tcccagccacggtgatag-3′), Pdk4 (forward primer: 5′-cgcttagtgaacactccttcg-3′ and reverse primer: 5′-cttctgggctcttctcatgg-3′), Cd36 (forward primer: 5′-cgttgtcatgatcctcatggt-3′ and reverse primer: 5′-acaggctgctcgggtc tat-3′), C/ebpα (forward primer: 5′-aaacaacgcaacgtggaga-3′ and reverse primer: 5′-gcggtcattgtcactggtc-3′), and Gapdh (glyceraldehyde 3-phosphate dehydrogenase; forward primer: 5′-tgtccgtcgtggatctgac-3′ and reverse primer: 5′-cctgcttcaccaccttc ttg-3′); designed by Universal ProbeLibrary Assay Design Center tool from Roche, Indianapolis, IN). The real-time PCR was performed with FastStart Universal Probe Master mix and specific primers and probes for each gene (Universal ProbeLibrary Single Probes No. 97 for Lrp1, No. 22 for Pdk4, No. 62 for Cd36, No. 67 for C/ebpα and No. 80 for Gapdh) in Roche Lightcycler 480 PCR machine. The reaction mixtures were incubated at 95 °C for 10 min followed by 55 cycles at 95 °C for 10 s and 60 °C for 30 s. GAPDH was used as the housekeeping gene.

**siRNA design and transient transfection.** The stealth siRNA duplexes were obtained from Life Technologies. The siRNA against mouse Pparγ is a duplex of 5′-ucaagggugccaguuucgauccgua-3′. The control siRNA is the Stealth RNAi negative control duplex (Cat. No. 12935-300) and was purchased from Life Technologies. The siRNAs were transfected into isolated Wt ECs according to our previous published protocol[11]. Briefly, for each sample, $2 \times 10^5$ ECs were transfected with 100 pmol siRNA. The experiments with siRNA-transfected ECs were performed 2 days later.

**Structure prediction.** The retrieved sequence of human LRP1β was used for the prediction of its secondary structure by SABLE server (http://sable.cchmc.org/)[38].

**Statistical analysis.** No statistical methods were used to predetermine the sample size. No randomization was used as all mice used were genetically defined, inbred mice. Data analysis for metabolic phenotype was performed in a blinded fashion. Statistical data were drawn from normally distributed group with similar variance between groups. All data presented in this study are representative results of at least three independent experiments. Data are shown as the mean ± s.e.m. 'n' represents the number of biological replicates. Differences were analysed with two-way analysis of variance and followed by a Fisher's least significant difference test unless otherwise specifically stated. Values of $P \leq 0.05$ were considered statistically significant.

**Data availability.** The data that support the findings of this study are available within the article, its Supplementary Information files and from the corresponding author upon reasonable request.

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

## Acknowledgements

We thank the Baylor Phenotyping Core, Optical Imaging and Vital Microscopy Core and Histology Core Laboratories for their help. This work was supported by NIH R01s HL122736 (to L.X.), HL112890 and HL061656 (to X.P.).

## Author contributions

H.M., C.M.B., C.P., L.X. and X.P. conceived the research and designed the experiments. H.M., P.L., L.L. and X.P. performed the experiments. H.M., C.M.B., L.X. and X.P. wrote the paper. All authors discussed the results and commented on the manuscript.
