## [Peer review file · Nature Communications]

Reviewers' comments:

Reviewer #1 (Remarks to the Author):

While several studies have suggested a link between LRP1 expression and glucose metabolism and obesity, the mechanisms of this link are completely unknown. The current study provides a plausible mechanism. The authors report that endothelial LRP1 contributes to mammalian lipid metabolism, glucose homeostasis, and regulation of fat mass by generating an endothelial cell (EC) specific LRP1 deletion. The resulting Cre⁺/BMT mice have decreased levels of LDL, HDL, TG and cholesterol levels after hyperlipidemic stress when compared to Cre⁻/BMT. The study further reports that these events occur via the ability of the LRP1 intracellular domain to interact with PPAR γ in endothelial cells and functions as a transcriptional co-activator. Finally, the study shows that a VGGLL sequence located in a coil region of LRP1 β -ICD maybe the required motif to bind to LBD of PPAR γ and act as a transcriptional co-activator.

These are novel and highly important observations that provides compelling evidence. Overall, this appears to be a well conducted study with valid approaches and appropriate conclusions drawn from the large amount of in vivo and in vitro data presented. The statistical methods appear appropriate.

A few issues, however, do need to be addressed:

1. While many of the phenotypes in endothelial-specific LRP1 deletion mimic that of PPAR γ endothelial specific knockout mice, the endothelial specific PPAR γ deficient mice do not show a difference in weight gain when fed a Western diet. This should be discussed
2. The data in Fig 2a and 2d are puzzling. At week 0, the studies show an increase in LDL in Cre⁺/BMT mice, but no change in total cholesterol. The assays used to quantify HDL and LDL cholesterol are not the best assays available. FPLC profiles should be shown to identify the specific lipoprotein particles that are affected.
3. The decrease in TG (Fig 2c) in Cre⁺/BMT mice suggests increased lipolysis in these mice. The activity of lipoprotein lipase on the endothelium should be quantified.

Reviewer #2 (Remarks to the Author):

Mice with endothelial cell (EC)-specific LRP1 deletion displayed improved metabolic responses during high-fat diet induced obesity. Mechanistically, we discovered that the processed LRP1 intracellular domain interacts with PPAR γ , a central regulator of lipid and glucose metabolism, and acts as a co-activator. This study establishes a novel receptor-dependent regulatory mechanism for PPAR activation, and suggests that LRP1, a new "dual address" protein, regulates lipid metabolism by not only acting as an endocytic receptor but also directly participating in gene transcription. In addition, it reveals an under-appreciated functional role of endothelium in maintaining energy homeostasis.

1. Fairly well established models are used here in terms of floxed mice and Tie2-Cre excision but given the importance to the hypotheses under study, expression of LRP1 in the liver should be demonstrated before and after Tie2Cre excision. This is relevant given that the liver also has endothelial cells although the structure of these ECs differ from peripheral ECs. Similarly, the reconstitution of LRP1 expression in hematopoietic cells (spleen, leukocytes) but not endothelial cells should be demonstrated, even if shown solely in Supplement.

2. The changes in higher LDL and lower HDL before high fat diet is noted but not the decrease in triglycerides, which is also relevant. Under basal conditions and prior to high fat, mice with deleted LRP1 have lower LDL, HDL and TG but no change in total cholesterol. This warrants some consideration? What were the VLDL levels?

3. This increased physical activity, which is probably due to increased oxidative metabolism of triglyceride-derived fatty acids in heart and skeletal muscle, may explain the decreased triglyceride level in Cre+/BMT mice (Figure 2c)

Can the authors provide some data to backup the notion that increased fatty acid oxidation causes increased activity (as suggested in quoted text above)? Usually this is the other way around - increased activity results in increased fatty acid oxidation. The increased activity at night may reflect CNS effects.

4.endothelium plays a role in insulin homeostasis, likely indirectly through the regulation of weight gain, adiposity or other mechanisms.

It appears the authors are suggesting the role of the endothelium in insulin homeostasis is indirect and a result of the decreased weight but the writing is not clear and creates an impression of a more direct or important role for endothelial LRP1 on insulin homeostasis when it would have to be considered be a consequence of the improved weight. How does weight gain differ from adiposity?

5. The evidence that LRP1 is increasing PPAR γ activity can be bolstered by showing increased expression of canonical PPAR γ target genes rather than just relying on PPRE luciferase activity, especially using such high levels of pioglitazone.

6. The finding that LRP1 β also regulates the other 2 PPAR isoforms is notable. If this is the case, does LRP1 deletion result in phenotypes that are a consequence of PPAR α deletion, which has also been shown to be important in systemic responses? Or PPAR δ ?

7. The authors may want to revisit the extent to which the discussion (as well as comments made in discussing results) are purely speculative. One example of many:

All these data could be resulted from the direct involvement of endothelial cells in lipid and glucose metabolism. Another possibility is that the crosstalk between vascular endothelium and its neighboring lipogenic cells might be crucial for energy homeostasis. Some unknown secreted energy regulators or membrane protein-protein interactions might be involved in this crosstalk, which will become one of our future directions.

How LRP1 deletion in endothelial cells is resulting in increased activity remains unexplained and may need further validation given the importance to understanding the phenotype.

8. The notion of LRP1 β as a novel, essential activator of PPAR γ is novel and worthy of further exploration/discussion. How does this insight impact the numerous prior LRP1 reports, especially in regards to liver and fat? Do any of those examine action of PPAR γ agonists and does the agonist fail to work in that setting?

Reviewer #3 (Remarks to the Author):

Mao H et al study the role of endothelial LRP1 on metabolism using a combination of mouse genetic models and in vitro cellular models. They demonstrate that specific deletion of LRP1 in endothelial cells improves insulin sensitivity, glucose tolerance and energy expenditure while body weight, LDL and HDL levels decrease in mice fed a high fat diet. It is noteworthy that a similar phenotype has been observed on glucose tolerance, energy expenditure and body weight in adipocyte-specific LRP1 knock out mice (Hoffman SM JCI 2017). At the molecular level, Mao et al demonstrate that LRP1 directly interacts with the nuclear PPAR receptors and increases their transcriptional activity.

Although PPAR γ has been shown to enhance LRP1 promoter activity via a PPRE in hepatocytes (Kim HJ Metabolism 2014), the direct interaction between both factors has not been reported so far. Therefore, it seems that part of LRP1's effect is mediated by PPAR γ , although a proper demonstration of this model is lacking. In particular, it is unclear what part of the EC LRP1 KO phenotype is due to the lack of PPAR γ activation, vs activation of the other PPARs or even other mechanisms. The authors should at least show which part of the in vivo phenotype is mediated via the PPAR γ interaction, for instance by treating the mice with a glitazone (not pioglitazone which may also activate PPAR α).

In addition, the mechanism regulating the effect of LRP1 in HFD-exposed endothelial cells is incomplete. Indeed, the interaction of PPAR γ with LRP1 in endothelial cells does not explain why and how endothelial LRP1 improve insulin sensitivity, glucose tolerance... in mice fed a HFD. The authors should also demonstrate a role for this mechanism in the LRP1-dependent response to a HFD.

Thus, additional experiments should be performed to improve the whole message and strengthened conclusions.

Other comments

1. Figure 2m: As most effects are observed after High Fat Diet feeding, and to be able to compare data obtained in figure 2n-o, insulin tolerance tests should be performed in mice after HFD exposure.
2. Figure 2a&b: How was LDL and HDL measured? It is assumed that it is rather LDL-cholesterol and HDL-cholesterol? The values are highly unusual for normal mice, which usually carry almost no cholesterol in LDL. The authors should perform lipoprotein cholesterol-distribution analysis by chromatography.
3. Figure 3c: the authors demonstrate that LRP1 does not interact anymore with PPAR γ - Δ LBD. The deletion of PPAR γ LBD domain may impair proper conformation of the remaining protein. The interaction of the PPAR γ LBD with LRP1 (or LRP1 ICD domain) should be properly demonstrated.
4. Figure 4d: do LRP1 deficient endothelial cells still respond to PPAR γ agonists in this context?

5. In addition, figure 1 and figure 2 demonstrate an effect of endothelial LRP1 in HFD-fed mice. What are the underlying mechanisms? What is the effect of long exposure of palmitate, for instance, on PPAR γ target genes in both endothelial cell models?
6. Figure 4g-i: the authors claim that these data may explain why LDL-C levels are higher in figure 2a. However, it does not explain why LDL and HDL levels are lower in figure 2a-b. Similar experiments should be performed in endothelial cells pre-treated with fatty acids and/or on endothelial cells isolated from mice fed a HFD.
7. Does LRP1-ICD interact with PPAR γ -LBD on the promoter of PPAR γ target genes? This should be addressed by chromatin immunoprecipitation assays using the different constructs described in this study. Does pioglitazone influence LRP1 binding to PPAR γ in this context? What is the effect of fatty acid sensitization on LRP1/PPAR γ complex formation? On LRP1 recruitment to PPAR γ target genes?
8. The link between metabolic phenotype and the proposed mechanism is unclear. What is the effect of thiazolidinedione treatment on metabolic parameters, energy expenditure, insulin sensitivity, glucose tolerance in endothelial-specific LRP1 KO mice compared to control littermates?
9. Mao et al study LRP1 as a coactivator, whose activity is mediated by PPAR γ . To investigate such hypothesis, and to decipher the PPAR γ -dependent from the PPAR γ -independent activity, the authors should at least investigate the effect of LRP1 overexpression on cholesterol uptake (oxLDL loading) and PPAR γ target gene expression in PPAR γ -deficient or knock down endothelial cells.
10. The authors should provide additional controls for the in vivo experiments. Since Cre expression may be sometimes non-specific and even spurious, they should check whether the Cre is not active and has knocked-out LRP1 in other metabolic tissues. The authors should also show that the in vivo phenotype is not due to Cre expression in ECs by showing that the Cre⁺ and Cre⁻ mice (in the background of wt LRP1) display similar phenotypes at baseline and after HFD.

Minor comment

1. Interestingly, leptin levels decreased in endothelial LRP1 KO mice fed a HFD compared to WT (Figure 2f) mice while the food intake is unchanged (Figure 2g), suggesting an increase of leptin sensitivity in endothelial LRP1 KO mice. It would be interesting to address such possibility.
2. As VO₂ is increased in endothelial LRP1 knock out mice (Figure 2h), it would be interesting to provide the RQ (Respiratory Quotient) to get further insights about the substrate (lipid vs glucose) used by both mouse lines.
3. Suppl. Information: the PPAR α agonist should be 'fenofibrate' instead of 'finofibrate'.

4. Figure legends are confusing, international nomenclature should be used to describe mouse line.
5. Statistics should be accurately described per panel.

Reviewer #1 (Remarks to the Author): *These are novel and highly important observations that provides compelling evidence. Overall, this appears to be a well conducted study with valid approaches and appropriate conclusions drawn from the large amount of in vivo and in vitro data presented. The statistical methods appear appropriate.*

1. While many of the phenotypes in endothelial-specific LRP1 deletion mimic that of PPARgamma endothelial specific knockout mice, the endothelial specific PPARgamma deficient mice do not show a difference in weight gain when fed a Western diet. This should be discussed.

We appreciate this great suggestion. The related discussion has been added into the Discussion section.

2. The data in Fig 2a and 2d are puzzling. At week 0, the studies show in increase LDL in Cre+/BMT mice, but no change in total cholesterol. The assays used to quantify HDL and LDL cholesterol are not the best assays available. FPLC profiles should be shown to identify the specific lipoprotein particles that are affected.

We really appreciate the Reviewer's great suggestion about FPLC profiles. We have performed FPLC profiles with serum from Cre-/BMT and Cre+/BMT mice. The data showed that at week 0, even before high-fat diet feeding, HDL-cholesterol, triglyceride (TG) levels were lower in Cre+/BMT mice than in Cre-/BMT control mice; however, VLDL-cholesterol and LDL-cholesterol levels were higher in Cre+/BMT mice than Cre-/BMT control mice (Supplementary Fig. 2a-d). Our FPLC data is consistent with the colorimetric assays performed for data shown in Fig. 2a-d. Following high-fat diet feeding for 16 weeks, TG level was significantly lower in Cre+/BMT mice than that in Cre-/BMT control mice, which was correlated to the dramatically decreased VLDL level in Cre+/BMT mice. LDL-cholesterol, HDL-cholesterol and total cholesterol were also significantly decreased in Cre+/BMT mice, compared to Cre-/BMT, following high-fat diet feeding. Taken all together, it suggests that endothelial LRP1 plays distinctive roles in lipid metabolism at the physiological condition and in response to hyperlipidemia stress.

3. The decrease in TG (Fig 2c) in Cre+/BMT mice suggests increased lipolysis in these mice. The activity of lipoprotein lipase on the endothelium should be quantified.

We really appreciate the great idea that the Reviewer provided to us. We have performed the LPL activity assay with mouse serum samples. The data was shown in Supplementary Fig. 2e. At week 0 and week 16, the LPL activity was indeed increased in Cre+/BMT mice, compared to Cre-/BMT control mice. This elevated LPL activity is inversely correlated with the reduction of serum TG level (Fig. 2c) in Cre+/BMT mice comparing to Cre-/BMT control mice, suggesting that LRP1 depletion in endothelial cells resulted in increased lipolysis. However, we did not detect dramatic changes in endothelial lipase mRNA levels in endothelial LRP1 knockout mice, compared to wild type endothelial cells (data not shown). The specific roles of different lipases including hepatic lipase, lipoprotein lipase and endothelial lipase in endothelial LRP1-mediated metabolic phenotypes still need more careful studies and will become one of our future research directions. We have incorporated these data in the Results and Supplemental Information sections.

Reviewer #2 (Remarks to the Author):

1. Fairly well established models are used here in terms of floxed mice and Tie2-Cre excision but given the importance to the hypotheses under study, expression of LRP1 in the liver should be demonstrated before and after Tie2Cre excision. This is relevant given that the liver also has endothelial cells although the structure of these ECs differ from peripheral ECs. Similarly, the reconstitution of LRP1 expression in

hematopoietic cells (spleen, leukocytes) but not endothelial cells should be demonstrated, even if shown solely in Supplement.

We thank the Reviewer for this great suggestion. We have performed immunostaining for liver sinusoidal endothelial cells of Cre-/BMT and Cre+/BMT mice. The immunostaining intensity of liver sinusoidal endothelial cells for Cre+/BMT is significantly decreased compared to Cre-/BMT control mice, suggesting knockout of LRP1 in liver sinusoidal endothelial cells (Supplementary Fig. 1e). In addition, we measured LRP1 mRNA levels by using real-time PCR assays for isolated endothelial cells, leukocytes, spleen lymphocytes, adipocytes, liver, kidney and heart of Cre- and Cre+ mice before and after bone marrow transplantation (Supplementary Fig. 1a, b). The mRNA levels of LRP1 were dramatically decreased in endothelial cells, leukocytes and lymphocytes in Cre+ mice, but not in adipocytes, liver, kidney and heart, compare to Cre- mice before bone marrow transplantation. After bone marrow transplantation, the mRNA levels of LRP1 in leukocytes and lymphocytes of Cre+/BMT mice were recovered back to similar levels as that of Cre-/BMT mice. However, the LRP1 mRNA level in endothelial cells was still significantly lower in Cre+/BMT mice than Cre-/BMT mice, suggesting that BMT resulted in the recovery of LRP1 expression in hematopoietic cells and the Cre+/BMT mice displayed a specific depletion of LRP1 in endothelial cells. We have incorporated these data in the Results and Supplemental Information sections.

2. The changes in higher LDL and lower HDL before high fat diet is noted but not the decrease in triglycerides, which is also relevant. Under basal conditions and prior to high fat, mice with deleted LRP1 have lower LDL, HDL and TG but no change in total cholesterol. This warrants some consideration? What were the VLDL levels?

The Reviewer has made a great point. As Reviewers suggested, we performed additional FPLC profiling assays to determine the specific lipoprotein particles. Our data based on both lipid colorimetric assays (Fig. 2a-d) and FPLC analysis (Supplementary Fig. 2a-d) suggest that, under basal conditions and prior to high-fat diet feeding, VLDL-cholesterol and LDL-cholesterol levels were higher, while HDL-cholesterol level is lower in Cre+/BMT mice, compared to Cre-/BMT control mice. This might explain why no significant change in total cholesterol level was observed between Cre+/BMT and Cre-/BMT mice. Our FPLC analysis also confirmed that triglyceride levels were decreased before and after high-fat diet in Cre+/BMT mice, compared to Cre-/BMT mice. Next, we performed the LPL activity assay with mouse serum samples. At week 0 and week 16, the LPL activity was indeed increased in Cre+/BMT mice, compared to Cre-/BMT control mice (Supplementary Fig. 2e). This elevated LPL activity is inversely correlated with the reduction of serum TG level (Fig. 2c) in Cre+/BMT mice comparing to Cre-/BMT control mice, suggesting that LRP1 depletion in endothelial cells resulted in increased lipolysis. We have updated the Results and Discussions sections to reflect these findings.

3. This increased physical activity, which is probably due to increased oxidative metabolism of triglyceride-derived fatty acids in heart and skeletal muscle, may explain the decreased triglyceride level in Cre+/BMT mice (Fig. 2c). Can the authors provide some data to backup the notion that increased fatty acid oxidation causes increased activity (as suggested in quoted text above)? Usually this is the other way around - increased activity results in increased fatty acid oxidation. The increased activity at night may reflect CNS effects.

We really appreciate the Reviewer's thoughtful comments and want to apologize for this misleading discussion. We have performed more thorough literature searches and totally agree with the Reviewer that increased activity usually results in increased fatty acid oxidation. This part has been corrected accordingly and updated in the Results section.

4. ...endothelium plays a role in insulin homeostasis, likely indirectly through the regulation of weight gain, adiposity or other mechanisms. It appears the authors are suggesting the role of the endothelium in insulin homeostasis is indirect and a result of the decreased weight but the writing is not clear and creates an impression of a more direct or important role for endothelial LRP1 on insulin homeostasis when it would have to be considered be a consequence of the improved weight. How does weight gain differ from adiposity?

Thank you for this great comment as this was not clear in the previous version of the manuscript. Yes, our data suggest that the role of the endothelium in insulin homeostasis is likely indirectly through the regulation of weight gain or other mechanisms. These related sentences have been modified to reflect our thoughts more clearly. Weight gain in response to high-fat feeding is due to the increase of adiposity and liver weight, based on our data in Figure 1. I hope that these changes have addressed your concern.

5. The evidence that LRP1 is increasing PPAR γ activity can be bolstered by showing increased expression of canonical PPAR γ target genes rather than just relying on PPRE luciferase activity, especially using such high levels of pioglitazone.

We highly appreciate the Reviewer's great comments. We have analyzed the expression of canonical PPAR γ target genes in MEFs (Fig. 3g) and endothelial cells (Fig. 4d-f, Supplementary Fig. 6a-d). In LRP1 knockout MEFs, PDK4 mRNA levels were decreased at basal condition and in response to pioglitazone treatment (Fig. 3g). In LRP1 knockout endothelial cells, there were much lower mRNA and protein levels of PPAR γ target genes CD36, PDK4 and C/EBP α , compared to Cre- control cells (Fig. 4d-f). We also performed additional experiments to study how LRP1 activates PPAR γ . First, we tested whether LRP1 regulates PPAR γ target gene expression by knocking down PPAR γ . As expected, treatments of overexpressed LRP1 β , pioglitazone or both increased mRNA levels of PPAR γ target genes. However, in PPAR γ knockdown endothelial cells, these increases were all inhibited (Supplementary Fig. 6c). In addition, we tested the effects of different agonists including thiazolidinediones (pioglitazone, ciglitazone, rosiglitazone and troglitazone) and palmitic acids on PPAR γ target gene expression. As expected, these agonists increased mRNA levels of PPAR γ target genes such as CD36, PDK4 and C/EBP α . However, these increases were inhibited in LRP1 depleted endothelial cells (Supplementary Fig. 6d). Next, we performed chromatin immunoprecipitation (ChIP) assays to determine whether LRP1 is associated with the promoter of PPAR γ target gene PDK4. Excitingly, the promoter sequence of PDK4 was detected in LRP1 bound chromatin complex, but not in control-IgG bound complex (Supplementary Fig. 3a), suggesting that LRP1 is associated with the PDK4 promoter. Our immunoprecipitation experiments also demonstrated that the association of LRP1 and PPAR γ was mildly increased by pioglitazone treatment (Supplementary Fig. 3b).

As the Reviewer commented, the *in vivo* luminescent imaging studies with PPRE-luc⁺ reporter mice were performed with a high dose of pioglitazone (150 mg/kg). Therefore, we repeated this experiments with much lower dose of pioglitazone (50 mg/kg). As expected, we still observed very strong luminescent signals in CAG-Cre⁻;PPRE-luc⁺ mice (revised Fig. 3f). However, the luminescent signals in CAG-Cre⁺;PPRE-luc⁺ mice were decreased at basal condition and in response to pioglitazone treatment. Based on our data from *in vitro* and *in vivo* reporter assays, real-time PCR and Western blotting of PPAR γ target genes, ChIP and co-immunoprecipitation assays, we conclude that LRP1 is required for ligand-dependent PPAR γ activation and its target gene induction. We have updated these results in the Results section accordingly.

6. The finding that LRP1 β also regulates the other 2 PPAR isoforms is notable. If this is the case, does LRP1 deletion result in phenotypes that are a consequence of PPAR α deletion, which has also been shown to be important in systemic responses? Or PPAR δ ?

We appreciate this suggestion to determine whether other 2 PPAR isoforms were mediators for LRP1-dependent metabolic responses. Our data demonstrate that LRP1 β also binds to PPAR α and PPAR β/δ and regulates their transcriptional activities (Supplementary Fig. 4). It is likely that the changes in PPAR α and PPAR β/δ activity in response to LRP1 deletion could also contribute to the improved systemic metabolic responses. We mainly study PPAR γ since there is a well-performed report supporting a pivotal role of endothelial PPAR γ in metabolic responses¹. However, we do not and should not exclude the possible roles of other PPARs, which are beyond the scope of our current investigation but will surely become one of our future directions.

Nevertheless, our data suggest that PPAR γ activity plays a role in LRP1-dependent metabolic responses. To further confirm the regulatory role of PPAR γ , we have performed additional experiments to test whether PPAR γ is required for the LRP1-dependent regulation of CD36, PDK4 and C/EBP α induction by using PPAR γ specific siRNAs. Our results show that overexpression of LRP1 β , treating endothelial cells with PPAR γ agonist pioglitazone, or both resulted in increases in mRNA levels of PPAR γ target genes CD36, PDK4 and C/EBP α . However, the expression levels of these genes were decreased significantly in PPAR γ knockdown cells (Supplementary Fig. 6c). The oxLDL loading assays demonstrate that PPAR γ knockdown significantly blocked cholesterol uptake that was induced by overexpression of LRP1 β , pioglitazone or both (Supplementary Fig. 6f). Taken together, our data provide strong evidence suggesting that LRP1-dependent PPAR γ activation contributes, at least partially, to the changes of metabolic responses in endothelial LRP1 knockout mice.

7. The authors may want to revisit the extent to which the discussion (as well as comments made in discussing results) are purely speculative.

One example of many:

All these data could be resulted from the direct involvement of endothelial cells in lipid and glucose metabolism. Another possibility is that the crosstalk between vascular endothelium and its neighboring lipogenic cells might be crucial for energy homeostasis. Some unknown secreted energy regulators or membrane protein-protein interactions might be involved in this crosstalk, which will become one of our future directions.

How LRP1 deletion in endothelial cells is resulting in increased activity remains unexplained and may need further validation given the importance to understanding the phenotype.

Thank you for these great comments. We have modified these discussion sections accordingly to avoid pure speculations.

In order to further understand how LRP1 deletion in endothelial cells is resulting in increased physical activity, we have performed the LPL activity assay with mouse serum samples. The data was shown in Supplementary Fig. 2e. At week 0 and week 16, the LPL activity was indeed increased in Cre⁺/BMT mice, compared to Cre⁻/BMT control mice. This elevated LPL activity is inversely correlated with the reduction of serum TG level (Fig. 2c) in Cre⁺/BMT mice comparing to Cre⁻/BMT control mice, suggesting that LRP1 depletion in endothelial cells resulted in increased lipolysis. This increased lipolysis (Supplementary Fig. 2e) is well correlated with the increased physical activity (Fig. 2h-j) and decreased triglyceride level in Cre⁺/BMT mice (Fig. 2c). We have incorporated these data in the Results and Supplemental Information sections.

8. The notion of LRP1 β as a novel, essential activator of PPAR γ is novel and worthy of further exploration/discussion. How does this insight impact the numerous prior LRP1 reports, especially in regards

to liver and fat? Do any of those examine action of PPAR γ agonists and does the agonist fail to work in that setting?

This suggestion is really great. We haven't found any examinations of PPAR γ agonists in prior LRP1 reports. Our observations provide new insights for LRP1's functional roles in metabolism, especially in regards to liver and fat. We have incorporated more detailed discussions in the Discussion section.

Reviewer #3 (Remarks to the Author):

*Although PPAR γ has been shown to enhance LRP1 promoter activity via a PPRE in hepatocytes (Kim HJ Metabolism 2014), the direct interaction between both factors has not been reported so far. Therefore, it seems that part of LRP1's effect is mediated by PPAR γ , although a proper demonstration of this model is lacking. In particular, it is unclear what part of the EC LRP1 KO phenotype is due to the lack of PPAR γ activation, vs activation of the other PPARs or even other mechanisms. The authors should at least show which part of the *in vivo* phenotype is mediated via the PPAR γ interaction, for instance by treating the mice with a glitazone (not pioglitazone which may also activate PPAR α). In addition, the mechanism regulating the effect of LRP1 in HFD-exposed endothelial cells is incomplete. Indeed, the interaction of PPAR γ with LRP1 in endothelial cells does not explain why and how endothelial LRP1 improve insulin sensitivity, glucose tolerance... in mice fed a HFD. The authors should also demonstrate a role for this mechanism in the LRP1-dependent response to a HFD. Thus, additional experiments should be performed to improve the whole message and strengthened conclusions.*

We totally agree with the Reviewer that further experiments need to be done to determine which part of the endothelial LRP1 knockout phenotype is due to the lack of PPAR γ activation. Therefore, we performed additional *in vitro* and *in vivo* experiments to dissect which part of the *in vivo* phenotype is mediated via the PPAR γ activation. We also determined whether LRP1/ PPAR γ -dependent signaling plays a role in LRP1-dependent responses following high-fat diet feeding, such as oxLDL-loaded cholesterol internalization in endothelial cells.

First, we tested whether LRP1 regulates PPAR γ target gene expression by knocking down PPAR γ or treating cells with PPAR γ agonists. As expected, treatments of overexpressed LRP1 β , pioglitazone or both increased mRNA levels of PPAR γ target genes CD36, PDK4 and C/EBP α . However, these increases were all inhibited in PPAR γ knockdown endothelial cells (Supplementary Fig. 6c). On the other hand, PPAR γ agonists including different thiazolidinediones (pioglitazone, ciglitazone, rosiglitazone and troglitazone) and palmitic acids increased mRNA levels of PPAR γ target genes. However, these increases were inhibited in LRP1 depleted endothelial cells (Supplementary Fig. 6d). We also isolated endothelial cells from endothelial LRP1 depleted mice following high-fat diet feeding for 9 weeks. Consistently, mRNA levels of CD36, PDK4 and C/EBP α were increased in Cre- control endothelial cells in response to high-fat diet feeding. However, these increases were abolished in LRP1 depleted Cre⁺ cells (Supplementary Fig. 6e).

In addition to these *in vitro* experiments, we also investigated whether PPAR γ agonists affect metabolic phenotypes of endothelial LRP1 depleted mice. Due to dual-specific effects of pioglitazone on both PPAR γ and PPAR α , we decided to test with both pioglitazone and another relatively specific PPAR γ agonist- rosiglitazone. Specifically, we analyzed metabolic parameters, energy expenditure, insulin and glucose tolerance responses in endothelial LRP1 knockout or their littermate control mice following the treatment of rosiglitazone or pioglitazone for 3 and 4 weeks, respectively. Our results demonstrate that in response to rosiglitazone and pioglitazone, most of metabolic phenotypes resulted from endothelial LRP1

depletion were still detected in these mice, compared to Cre-/BMT mice (Supplementary Fig. 7). Taken all together, our additional data provide further mechanistic support for our hypothesis that LRP1 is a co-activator of PPAR γ and required for its ligand-dependent target gene induction.

We also performed oxLDL loading assays with PPAR γ knockdown endothelial cells. Our results indicate that PPAR γ was required for cholesterol internalization induced by overexpression of LRP1, pioglitazone treatment or both (Supplementary Fig. 6f). In response to PPAR γ agonists pioglitazone and palmitic acids, cholesterol uptake was increased. However, these increases were blocked in LRP1-depleted endothelial cells (Fig. 4h, Supplementary Fig. 6g). This suggests that PPAR γ activity mediates LRP1-dependent cholesterol internalization. Last, we tested cholesterol uptake with endothelial cells isolated from 9-week-high-fat fed mice. We discovered that cholesterol uptake was significantly decreased in LRP1 knockout endothelial cells in control chow fed mice, compared to Cre- control cells (Supplementary Fig. 6h). However, very surprisingly, cholesterol uptake was increased dramatically in LRP1 knockout endothelial cells isolated from high-fat diet fed mice, compared to Cre- control endothelial cells isolated from high-fat diet fed mice, or LRP1 knockout endothelial cells isolated from control chow fed mice (Supplementary Fig. 6h). This increase, inversely correlated to decreased LDL-cholesterol level following high-fat diet feeding in endothelial LRP1 knockout mice (Fig. 2a, Supplementary Fig. 2a-b), could not be explained by the decreased induction of CD36 in the same cells (Supplementary Fig. 6e). It suggests that endothelial LRP1 plays an active role in LDL-cholesterol clearance at basal condition and in response to hyperlipidemia. CD36 is required for LRP1-mediated cholesterol internalization at basal condition. However, hyperlipidemia stress activates CD36-independent mediators for cholesterol internalization.

This, together with our other observations such as the potential roles of endothelial LRP1 in HDL and triglyceride homeostasis, insulin sensitivity and glucose tolerance will become our future research directions. Nevertheless, our current results strongly suggest that endothelial LRP1 plays a pivotal role in the regulation of metabolic homeostasis, at least partially, through the regulation of PPAR γ transcriptional activation.

Our data demonstrate that LRP1 β also binds to PPAR α and PPAR β/δ and regulates their transcriptional activities (Supplementary Fig. 5). It is likely that the changes in PPAR α and PPAR δ activity in response to LRP1 deletion could also contribute to the improved systemic metabolic responses. We study PPAR γ since there is a well-performed report supporting a pivotal role of endothelial PPAR γ in metabolic responses¹. However, we do not and should not exclude the possible roles of other PPARs, which are beyond the scope of our current investigation but will surely become one of our future directions.

1. Figure 2m: As most effects are observed after High Fat Diet feeding, and to be able to compare data obtained in figure 2n-o, insulin tolerance tests should be performed in mice after HFD exposure.

Thank you very much for this great suggestion. We added the data of insulin tolerance tests that we performed in Cre-/BMT and Cre+/BMT mice after high-fat diet feeding (Supplementary Fig. 2h). As mentioned in Fig. 2m, Cre+/BMT mice demonstrated improved insulin sensitivity compared to Cre-/BMT control mice. After high-fat diet feeding for 16 weeks (week 16), Cre+/BMT mice still demonstrated increased insulin sensitivity, compare to Cre-/BMT mice (Supplementary Fig. 2h).

2, Figure 2a&b: How was LDL and HDL measured? It is assumed that it is rather LDL-cholesterol and HDL-cholesterol? The values are highly unusual for normal mice, which usually carry almost no cholesterol in LDL. The authors should perform lipoprotein cholesterol-distribution analysis by chromatography.

Thank you very much for the suggestion. The LDL and HDL measurements were performed with colorimetric assays to quantify the cholesterol contents in LDL or HDL particles. As the Reviewers suggested, we have performed additional LDL, HDL measurements by using lipoprotein cholesterol-

distribution analysis with FPLC. Indeed, the numbers of HDL-cholesterol and LDL-cholesterol levels are lower than the readings with colorimetric assays (Supplementary Fig. 2a, b). However, we observed similar trends in LDL and HDL-cholesterol changes in Cre⁺/BMT mouse serum, compared to Cre⁻/BMT control before and after high-fat diet feeding. All these data suggest that LRP1 depletion in endothelial cells contributes to the regulation of lipid homeostasis.

3. Figure 3c: the authors demonstrate that LRP1 does not interact anymore with PPAR γ -ALBD. The deletion of PPAR γ LBD domain may impair proper conformation of the remaining protein. The interaction of the PPAR γ LBD with LRP1 (or LRP1 ICD domain) should be properly demonstrated.

We appreciate the Reviewer's careful evaluation about the experiments for LRP1 and PPAR γ interaction. We have demonstrated that the intracellular domain (ICD) of LRP1 β bound to PPAR γ by using GST pull-down assays with purified recombinant protein of LRP1-ICD (Fig. 3b). Actually we have also performed immunoprecipitation experiments to determine the interaction of purified PPAR γ LBD protein (commercially available) and overexpressed Flag-tagged LRP1 β protein. As expected, their interaction was observed (Fig. 3h). Together with the results with deletion mutations of PPAR γ (Fig. 3c), we conclude that LRP1 is associated with PPAR γ through the LBD domain of PPAR γ and ICD domain of LRP1. We hope these data have addressed the Reviewer's concerns.

4. Figure 4d: do LRP1 deficient endothelial cells still respond to PPAR γ agonists in this context?

The suggestion to examine the response upon PPAR γ 's agonists in LRP1 deficient endothelial cells is great. We checked the mRNA levels of PPAR γ target genes in LRP1 deficient endothelial cells by using different PPAR γ 's agonists. Our data demonstrate that PPAR γ 's agonists, including different thiazolidinediones (pioglitazone, ciglitazone, rosiglitazone and troglitazone) and palmitic acids, increased mRNA levels of PPAR γ target genes CD36, PDK4 and C/EBP α . However, these increases were inhibited in LRP1 depleted endothelial cells (Supplementary Fig. 6d). It suggests that LRP1, acting as PPAR γ co-activator, is required for PPAR γ activity at basal condition and in response to agonists. We have these data incorporated into the Results section accordingly.

5. In addition, figure 1 and figure 2 demonstrate an effect of endothelial LRP1 in HFD-fed mice. What are the underlying mechanisms? What is the effect of long exposure of palmitate, for instance, on PPAR γ target genes in both endothelial cell models?

We appreciate this great suggestion and thoughtful questions. In response to the reviewer's questions, we have fed our mice with high-fat diet for 9 weeks and isolated endothelial cells from these mice. Then we analyzed mRNA levels of PPAR γ 's target genes CD36, PDK4 and C/EBP α in the hyperlipidemia exposed endothelial cells. Our results demonstrate that long term hyperlipidemia resulted in increased expression of PPAR γ 's target genes CD36 and PDK4, and mildly, C/EBP α . However, these increases were inhibited in LRP1 depleted endothelial cells (Supplementary Fig. 6e). In addition, we treated isolated endothelial cells with palmitic acids for 24 hours and then measured mRNA levels of these PPAR γ 's target genes. Our results indicated that, similarly to the hyperlipidemia exposure of endothelial cells *in vivo*, long term exposure of palmitic acids led to increases in mRNA levels of PPAR γ 's target genes in Cre⁻ control endothelial cells. However, these increases were blocked in LRP1 depleted endothelial cells (Supplementary Fig. 6d). Taken all together, our data suggest that LRP1, acting as a co-activator of PPAR γ , is required for expression of PPAR γ 's target genes at basal condition and in response to high fat challenge.

As I mentioned in the answer to your first comment, to understand the effect of endothelial LRP1 in HFD fed mice, we have investigated whether PPAR γ agonists affect metabolic phenotypes of endothelial LRP1 depleted mice. Our results demonstrate that in response to rosiglitazone and pioglitazone, most of metabolic

phenotypes resulted from endothelial LRP1 depletion were still detected in these mice, compared to Cre-/BMT mice (Supplementary Fig. 7). We also performed oxLDL loading assays with PPAR γ knockdown endothelial cells. We demonstrated that PPAR γ was required for cholesterol internalization induced by overexpression of LRP1, pioglitazone treatment or both (Supplementary Fig. 6f). In response to PPAR γ agonists pioglitazone and palmitic acids, cholesterol uptake was increased. However, these increases were blocked in LRP1-depleted endothelial cells (Fig. 4h, Supplementary Fig. 6g). This suggests that PPAR γ activity mediates LRP1-dependent cholesterol internalization. Last, we tested cholesterol uptake with endothelial cells isolated from 9-week-high-fat diet fed mice. We discovered that cholesterol uptake was significantly decreased in LRP1 knockout endothelial cells in control chow fed mice, compared to Cre-control cells (Supplementary Fig. 6h). However, very surprisingly, cholesterol uptake was increased dramatically in LRP1 knockout endothelial cells isolated from high-fat diet fed mice, compared to Cre-control endothelial cells isolated from high-fat diet fed mice, or LRP1 knockout endothelial cells isolated from control chow fed mice (Supplementary Fig. 6h). This increase, inversely correlated to decreased LDL-cholesterol level following high-fat diet feeding in endothelial LRP1 knockout mice (Fig. 2a, Supplementary Fig. 2a-b), could not be explained by the decreased induction of CD36 in the same cells (Supplementary Fig. 6e). It suggests that endothelial LRP1 plays an active role in LDL-cholesterol clearance at basal condition and in response to hyperlipidemia. CD36 is required for LRP1-mediated cholesterol internalization at basal condition. However, hyperlipidemia stress activates CD36-independent mediators for cholesterol internalization.

This, together with our other observations such as the potential roles of endothelial LRP1 in HDL and triglyceride homeostasis, insulin sensitivity and glucose tolerance will become our future research directions. Nevertheless, our current results strongly suggest that endothelial LRP1 plays a pivotal role in the regulation of metabolic homeostasis, at least partially, through the regulation of PPAR γ transcriptional activation. We have included these data in the Results section.

6. Figure 4g-i: the authors claim that these data may explain why LDL-C levels are higher in figure 2a. However, it does not explain why LDL and HDL levels are lower in figure 2a-b. Similar experiments should be performed in endothelial cells pre-treated with fatty acids and/or on endothelial cells isolated from mice fed a HFD.

We appreciate the Reviewer's great suggestions. Yes, the data in Fig. 4g-i suggest a plausible molecular mechanism for the observed higher LDL-cholesterol level in Fig. 2a. To understand better why the LDL-cholesterol level was lower following high-fat diet feeding in Cre+/BMT mice than that in Cre-/BMT control mice, we have performed two additional experiments. First, the isolated LRP1 depleted endothelial cells and Cre- controls cells were treated with palmitic acids for 24 hours and the internalized cholesterol levels were measured following oxLDL loading. Indeed, palmitic acids treatment, which increased CD36 mRNA levels (Supplementary Fig. 6d), led to an increase in cholesterol internalization in Cre- endothelial cells. However, in LRP1 depleted endothelial cells, palmitic acids failed to increase CD36 mRNA level and cholesterol internalization (Supplementary Fig. 6d, 6g). We also fed both Cre+ and Cre- mice with high-fat diet for 9 weeks and then isolated their endothelial cells. We performed oxLDL loading assays and discovered that cholesterol internalization was inhibited in Cre- control endothelial cells that have been exposed to high-fat diet for 9 weeks. However, very surprisingly, cholesterol uptake was increased dramatically in LRP1 knockout endothelial cells isolated from high-fat diet fed mice, compared to Cre- control endothelial cells isolated from high-fat diet fed mice, or LRP1 knockout endothelial cells isolated from control chow fed mice (Supplementary Fig. 6h). This increase, inversely correlated with the decreased LDL cholesterol level in Cre+/BMT mice following high-fat diet feeding for 16 weeks (Fig. 2a), could not be explained by the decreased induction of CD36 in the same cells (Supplementary Fig. 6e). It suggests that endothelial LRP1 plays an active role in LDL-cholesterol clearance at basal condition and in response to hyperlipidemia. CD36 is required for LRP1-mediated cholesterol internalization at basal condition. However, hyperlipidemia stress

activates CD36-independent mediators for cholesterol internalization. As the Reviewer mentioned, we also observed the decrease of HDL-cholesterol level before and after high-fat diet feeding, which is similar to that phenotype with liver-specific LRP1 knockout mice^{2,3}. However, whether and how endothelial LRP1 plays a role in HDL metabolism remains mystery and will become one of our future research directions. We have included these data and thoughts in the Results and Discussion sections.

7. Does LRP1-ICD interact with PPAR γ -LBD on the promoter of PPAR γ target genes? This should be addressed by chromatin immunoprecipitation assays using the different constructs described in this study. Does pioglitazone influence LRP1 binding to PPAR γ in this context? What is the effect of fatty acid sensitization on LRP1/PPAR γ complex formation? On LRP1 recruitment to PPAR γ target genes?

We appreciate these thoughtful questions. As the Reviewer suggested, we have performed chromatin immunoprecipitation (ChIP) assays and included the data in our revised manuscript. The stable LRP1 β transfected HEK293 cells were used to make soluble chromatin. Following the immunoprecipitation with the chromatin fraction by using LRP1 β antibody or rabbit IgG as a control, the promoter of PPAR γ target gene PDK4 was detected by PCR with its specific primers. Our ChIP data demonstrate that the PDK4 promoter was detected in LRP1 β associated complex, but not in the rabbit IgG associated complex. The association was still observed upon treatments of pioglitazone or palmitic acids (Supplementary Fig. 3a). We also performed immunoprecipitation experiments to test whether pioglitazone influences LRP1 binding to PPAR γ . In mouse endothelial cells, the binding of LRP1 and PPAR γ was mildly increased along the treatments of pioglitazone for 5 to 15 minutes (Supplementary Fig. 3b). We have incorporated these data into the Results section.

8. The link between metabolic phenotype and the proposed mechanism is unclear. What is the effect of thiazolidinedione treatment on metabolic parameters, energy expenditure, insulin sensitivity, glucose tolerance in endothelial-specific LRP1 KO mice compared to control littermates?

We appreciate this great suggestion. As the Reviewer suggested, we have performed further analysis with metabolic parameters, energy expenditure, insulin sensitivity and glucose tolerance following i.p. injection of rosiglitazone or pioglitazone for 3 or 4 weeks, respectively. As shown in Supplementary Fig. 7, following rosiglitazone or pioglitazone injections, endothelial-specific LRP1 knockout mice still demonstrated similar changes in metabolic parameters, increased physical activity, improved insulin sensitivity and glucose tolerance responses. It suggests that LRP1 depletion in endothelial cells leads to improved metabolic responses at basal condition and in response to PPAR γ activation by rosiglitazone or pioglitazone. These data further support our hypothesis that LRP1 is a co-activator of PPAR γ and is required for its ligand-dependent activation.

9. Mao et al study LRP1 as a coactivator, whose activity is mediated by PPAR γ . To investigate such hypothesis, and to decipher the PPAR γ -dependent from the PPAR γ -independent activity, the authors should at least investigate the effect of LRP1 overexpression on cholesterol uptake (oxLDL loading) and PPAR γ target gene expression in PPAR γ -deficient or knock down endothelial cells.

This is a great suggestion. We have performed additional experiments to investigate how LRP1 overexpression affects PPAR γ target gene expression and cholesterol uptake in PPAR γ knock down endothelial cells. Our results demonstrate that, treatments of overexpressed LRP1 β , pioglitazone or both increased mRNA levels of PPAR γ target genes. However, these increases were all inhibited in PPAR γ knockdown endothelial cells (Supplementary Fig. 6c). We also observed that PPAR γ was required for cholesterol internalization induced by overexpression of LRP1, pioglitazone treatment or both (Supplementary Fig. 6f). Taken together, our results further demonstrate that LRP1 β is required for the ligand-dependent PPAR γ target gene induction and cholesterol uptake.

10. The authors should provide additional controls for the *in vivo* experiments. Since Cre expression may be sometimes non-specific and even spurious, they should check whether the Cre is not active and has knocked-out LRP1 in other metabolic tissues. The authors should also show that the *in vivo* phenotype is not due to Cre expression in ECs by showing that the Cre⁺ and Cre⁻ mice (in the background of wt LRP1) display similar phenotypes at baseline and after HFD.

We greatly appreciate the Reviewer's careful evaluation of our *in vivo* data and totally understand the concerns of the Reviewer. We have performed immunostaining for liver sinusoidal endothelial cells of Cre⁻/BMT and Cre⁺/BMT mice. The immunostaining intensity of liver sinusoidal endothelial cells for Cre⁺/BMT is significantly decreased compared to Cre⁻/BMT control mice, suggesting knockout of LRP1 in liver sinusoidal endothelial cells (Supplementary Fig. 1e). As the Reviewers suggested, we also measured LRP1 mRNA levels by using real-time PCR assays for isolated endothelial cells, leukocytes, lymphocytes, adipocytes, liver, kidney and heart of Cre⁻ and Cre⁺ mice before and after bone marrow transplantation (Supplementary Fig. 1a, b). The mRNA levels of LRP1 were dramatically decreased in endothelial cells, leukocytes and lymphocytes in Cre⁺ mice, but not in adipocytes, liver, kidney and heart, compare to Cre⁻ mice (Supplementary Fig. 1a). After bone marrow transplantation, the mRNA levels of LRP1 in leukocytes and lymphocytes of Cre⁺/BMT mice were increased to similar levels as that of Cre⁻/BMT mice. However, the LRP1 mRNA level in endothelial cells was still significantly lower in Cre⁺/BMT mice (Supplementary Fig. 1b). These data suggest that LRP1 expression was recovered in hematopoietic cells and LRP1 in endothelial cells was specifically depleted after bone marrow transplantation.

Regarding to the concern about the non-specific effects of Tie2Cre transgene, before we started these *in vivo* experiments, we have searched literatures about the Tie2Cre transgenic mice and found from many reports that no obvious changes in metabolic responses were observed in Tie2Cre transgenic mice or Tie2Cre-mediated knockout mice. For example, studies have been done to compare the differences of Tie2Cre transgenic mice and wild type mice in metabolic responses and no obvious differences in body weight and blood glucose and insulin levels and other metabolic responses were observed⁴. Another study has tracked metabolic responses until mice were one year old and found that the Tie2Cre-mediated insulin receptor knockout mice did not change whole-body glucose tolerance, insulin sensitivity and plasma lipids compared to the intact insulin receptor control mice⁵. Therefore, we hope that these reports have addressed the Reviewer's concerns.

Minor comment

1. Interestingly, leptin levels decreased in endothelial LRP1 KO mice fed a HFD compared to WT (Figure 2f) mice while the food intake is unchanged (Figure 2g), suggesting an increase of leptin sensitivity in endothelial LRP1 KO mice. It would be interesting to address such possibility.

This is a great suggestion. We have included this point in the Results section.

2. As VO₂ is increased in endothelial LRP1 knock out mice (Figure 2h), it would be interesting to provide the RQ (Respiratory Quotient) to get further insights about the substrate (lipid vs glucose) used by both mouse lines.

Thanks for this great comment. We have calculated RER (respiratory exchange ratio), an estimate of RQ, for the data of indirect calorimetry studies. There were no significant differences in RER values of Cre⁺/BMT and Cre⁻/BMT mice (Supplementary Fig. 2g). Both values were at 0.91~0.95 and close to 1.0, suggesting that carbohydrate was the primary fuel source for both endothelial LRP1 knockout and control mice.

3. *Suppl. Information: the PPAR α agonist should be 'fenofibrate' instead of 'finofibrate'.*

Thank the Reviewer for the careful reading. We would like to apologize for our oversight. 'finofibrate' was replaced by 'fenofibrate' in our revised manuscript.

4. *Figure legends are confusing, international nomenclature should be used to describe mouse line.*

We have corrected the names of the mouse lines in the figure legends.

5. *Statistics should be accurately described per panel.*

As the Reviewer suggested, the statistics were updated in the supplemental Methods and the related figure legends.

References

1. Kanda, T., *et al.* PPAR γ in the endothelium regulates metabolic responses to high-fat diet in mice. *J Clin Invest* **119**, 110-124 (2009).
2. Basford, J.E., *et al.* Hepatic deficiency of low density lipoprotein receptor-related protein-1 reduces high density lipoprotein secretion and plasma levels in mice. *J Biol Chem* **286**, 13079-13087 (2011).
3. Rohlmann, A., Gotthardt, M., Hammer, R.E. & Herz, J. Inducible inactivation of hepatic LRP gene by cre-mediated recombination confirms role of LRP in clearance of chylomicron remnants. *J Clin Invest* **101**, 689-695 (1998).
4. Vicent, D., *et al.* The role of endothelial insulin signaling in the regulation of vascular tone and insulin resistance. *J Clin Invest* **111**, 1373-1380 (2003).
5. Rask-Madsen, C., *et al.* Loss of insulin signaling in vascular endothelial cells accelerates atherosclerosis in apolipoprotein E null mice. *Cell Metab* **11**, 379-389 (2010).

Reviewers' comments:

Reviewer #1 (Remarks to the Author):

In this revised manuscript, the authors have performed a number of additional experiments that provide substantial data to address the prior critique. In my view, the new experiments appropriately address the critiques. Overall, this is an important study that for the first time reveals an important role for LRP1 in regulating PPAR activation. The experiments appear carefully done and statistical analysis of the data is appropriate.

Reviewer #3 (Remarks to the Author):

Mao H et al demonstrate in this study that endothelial LRP1 potentiates the role of PPAR γ on lipoprotein transport and insulin sensitivity. The authors have elegantly and properly addressed our main concerns.

However, the authors should modify the legends according to journal guidelines by giving the exact number of mice per group and not a rank as it is currently stated; accurately describe replicates (whether they are technical or biological replicates, whether it is a representative experiment of at least 3 independent experiments).

The authors should also provide another negative DNA region control for the ChIP experiment, which is an essential control showing the DNA region specificity. Authors may use primers designed in the vicinity of LRP1 binding site in the PDK4 promoter (between 1-5 kb from the binding site). In addition, the relative binding of LRP1 between different experimental conditions should be accurately quantified using ChIP-qPCR but ChIP-PCR, otherwise conclusions, which are only based on PCR product intensity, are overstated. Finally, does the LRP1 mutant still bind to the PDK4 promoter?

Reviewer #3 (Remarks to the Author):

Mao H et al demonstrate in this study that endothelial LRP1 potentiates the role of PPAR γ on lipoprotein transport and insulin sensitivity. The authors have elegantly and properly addressed our main concerns.

However, the authors should modify the legends according to journal guidelines by giving the exact number of mice per group and not a rank as it is currently stated; accurately describe replicates (whether they are technical or biological replicates, whether it is a representative experiment of at least 3 independent experiments).

This is a great comment! We have carefully reviewed our original manuscript and updated the information about mouse numbers per group in the Figure Legends. All the experiments have been performed for at least three different biological replicates and all the figures are representative experiments of at least 3 independent experiments. The related information has been updated in the Methods section.

The authors should also provide another negative DNA region control for the ChIP experiment, which is an essential control showing the DNA region specificity. Authors may use primers designed in the vicinity of LRP1 binding site in the PDK4 promoter (between 1-5 kb from the binding site). In addition, the relative binding of LRP1 between different experimental conditions should be accurately quantified using ChIP-qPCR but ChIP-PCR, otherwise conclusions, which are only based on PCR product intensity, are overstated. Finally, does the LRP1 mutant still bind to the PDK4 promoter?

Thank you very much for this thoughtful comment. As the Reviewer suggested, we have performed chromatin immunoprecipitation (ChIP) assays with additional controls and quantitative PCR (qPCR). We used the specific primers of PDK4 promoter to determine whether LRP1 is associated with the promoter of PDK4. We also designed negative DNA region control primers that are in the vicinity of LRP1 binding site (~3 kb far) in the PDK4 promoter. Our new

data demonstrate that LRP1 is specifically in a complex with the promoter of PDK4 and this association is increased upon the treatment of pioglitazone or palmitic acids. The original data has been replaced with this new one in Supplemental Figure 3a.

We also performed additional ChIP- qPCR assays. HEK293 cells were transfected with Flag-tagged LRP1 β wild-type (Wt) or mutant (Mut) constructs. Chromatin immunoprecipitation was performed by using Flag antibody or mouse IgG as a negative control, followed by qPCR for the PDK4 promoter. We observed that LRP1 β -Wt was in a complex with the PDK4 promoter. However, this association was significantly decreased for LRP1 β -Mut. The data has been updated in the revised Supplemental Figure 3b.

Taken all together, our additional data provide further mechanistic support for our hypothesis that LRP1 is a co-activator of PPAR γ and required for its ligand-dependent activation and target gene induction.